# Computational models of episodic-like memory in food-caching birds

**Johanni Brea** [1,2] ✉, **Nicola S. Clayton**[3] **& Wulfram Gerstner**[1,2]

Birds of the crow family adapt food-caching strategies to anticipated needs at the time of cache recovery and rely on memory of the what, where and when of previous caching events to recover their hidden food. It is unclear if this behavior can be explained by simple associative learning or if it relies on higher cognitive processes like mental time-travel. We present a computational model and propose a neural implementation of food-caching behavior. The model has hunger variables for motivational control, reward-modulated update of retrieval and caching policies and an associative neural network for remembering caching events with a memory consolidation mechanism for flexible decoding of the age of a memory. Our methodology of formalizing experimental protocols is transferable to other domains and facilitates model evaluation and experiment design. Here, we show that memory-augmented, associative reinforcement learning without mental time-travel is sufficient to explain the results of 28 behavioral experiments with food-caching birds.

Food-caching birds of the crow family (*Corvidae*) have been proposed as animal models for cognitive neuroscience, because of their remarkably complex cognition[1]. In the wild, nutcrackers and jays cache food items like nuts or insects in thousands of small cracks or in loose soil, for hours, days or months. Recovery of their own caches is highly probable (50–99%), dependent on visual cues, independent of olfactory cues and inconsistent with random search at preferred locations[2]. In laboratory experiments (Fig. 1A), jays rely on episodic-like memories of what they cached where and when (henceforth called 'memory experiments'[3–7]) and adapt their caching strategy to anticipated future needs ('planning experiments'[8–12]), among other behavior[1].

The interpretation of the planning experiments is controversial. Are they evidence for 'mental time-travel'[10–12], a characteristic feature of human behavior[13]? Or are they explained by a mnemonic-associative recall of previous actions at the time of cache recovery[8,9,14]? A resolution of this controversy would determine those aspects of cognition for which corvids are representative animal models.

To formalize different interpretations, we turn to reinforcement learning[15,16], a computational framework to describe behavioral experiments. Reinforcement learning models are distinguished by how action selection depends on past experiences. First, in a model-free approach, action selection depends on a policy, whose parameters (e.g. synaptic weights) are directly changed by experience. Memory about past events is stored implicitly in these policy parameters. Second, in a model-based approach, past experiences are used to build a model of the world in the form of causal information like "doing A in state B has consequences C with probability P". This model can be used for 'planning at decision time' or to update a model-free policy[17]. Memory about past events is stored in aggregated form in the model of the world. Third, in a buffer-based approach, the sequence of past experiences is explicitly stored in a memory buffer. Using 'experience-replay', this buffer can be used to repeat actions with successful outcome or to update a world-model or a model-free policy.

Model-free reinforcement learning is usually associated with learning of habits. Often this simple form of learning is sufficient to explain behavior: tool-use experiments with corvids and apes can be explained by model-free reinforcement learning[18] and the mnemonic-associative account[14] of the planning experiments resembles model-free reinforcement learning[19]. On the other hand, there is evidence for simple versions of model-based reinforcement learning in different species[20,21] and processes like planning or mental time-travel require a model- or buffer-based approach[16,22].

[1]School of Computer and Communication Science, École Polytechnique Fédérale de Lausanne, Lausanne, Switzerland. [2]School of Life Science, École Polytechnique Fédérale de Lausanne, Lausanne, Switzerland. [3]Department of Psychology, University of Cambridge, Cambridge, UK. ✉e-mail: johanni.brea@epfl.ch

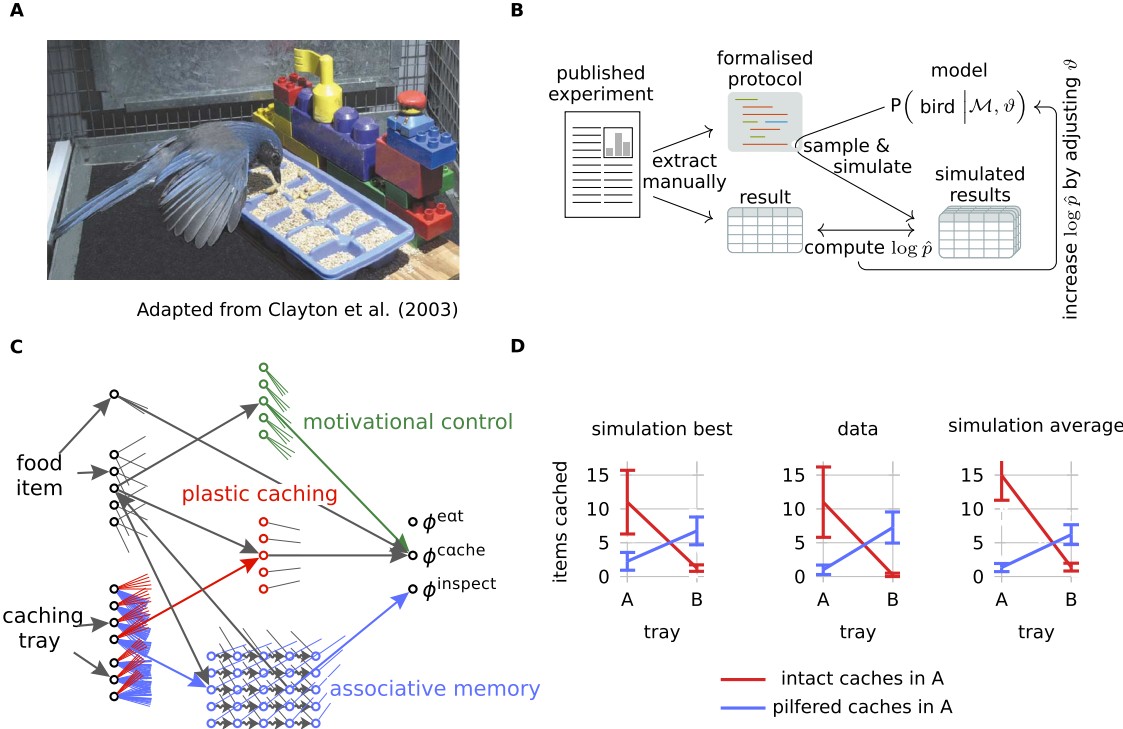

**Fig. 1 | Computational modeling approach. A** In the experiments, jays can eat different items of food, cache them in visuospatially distinct sites (ice cube trays with different arrangements of LEGO Duplo blocks) or inspect their own caches for available food. With different feeding schedules and manipulations of the cached food items the experimenter can change the birds' motivational states and caching experiences. Adapted from Clayton, N., Bussey, T. & Dickinson, A. Can animals recall the past and plan for the future?. Nat Rev Neurosci 4, 685-691 (2003). https://doi.org/10.1038/nrn1180, Springer Nature Limited. **B** For 28 published experiments we formalized the experimental protocol in a domain-specific language and extracted all measured quantities from text and figures. Our models (top right) describe populations of simulated birds as distributions over dynamical systems. Simulated birds are sampled from these populations to participate in simulated runs of the experimental protocols and produce simulated results. We compute the approximate log-likelihood $\log \hat{p}$ by comparing simulated with actual results (Methods). The hyperparameters $\vartheta$ that determine the mean behavior and the inter-

individual differences are adjusted to maximize $\log \hat{p}$. **C** In the Plastic Caching Model the preferences of eating $\phi^{\mathrm{eat}}$, caching $\phi^{\mathrm{cache}}$, and inspecting $\phi^{\mathrm{inspect}}$ depend on the neural activities and synaptic strengths in a structured neural network featuring motivational control (green), plastic caching weights (red) and an associative memory with systems consolidation (blue). Only some connections are fully drawn; short outgoing lines indicate connections that are suppressed in this figure. The hyperparameters $\vartheta$ set initial values of the weights and the speed of learning.
**D** Example experiment 'deKort07 exp4a' (see Figs. S9–S168 for other experiments and all 78 comparison figures per model). Birds that experienced pilfered caches in tray A (blue) stopped caching food in their previously preferred tray A, whereas birds that successfully retrieved their caches in A continued to cache preferentially in A (center, error bars = SEM, $n = 4$ birds per group). These experimental results are reproduced in simulations with the Plastic Caching Model (left: best out of $10^5$ simulations, error bars = SEM). Also on average (right), the simulated results match qualitatively the experimental data (error bars = average SEM).

Model-free, model-based or buffer-based approaches allow reinforcement learning agents to adapt their policies to the specificities of the environment. In addition to this kind of memory, agents need dedicated memory if they act in a partially observable domain, where perceptions inform only partially about the full state of the environment[23]. Because cached food is hidden, food-caching animals need this additional kind of memory to remember their food caches. Finally, the birds' behavior is affected by internal motivational states, like the level of hunger for each type of food[11,12,24]. Hence, in a reinforcement learning model of food-caching animals, policy and reward are functions of external stimuli (e.g. visual cues), internal motivational states (e.g. hunger) and memory of food caches.

Here, we ask whether the birds' food-caching behavior can or cannot be explained by established concepts like reward-modulated synaptic plasticity and associative memory networks[25]. To answer this question, we translated different explanations of food-caching behavior into computational models of neural circuits of memory and decision making, formalized experimental protocols in a domain-specific language (Figs. 1B and Fig. S5) and compared simulated to measured results extracted from 28 published experiments (1568 data points).

## Results

Each computational model specifies a population of simulated birds, with mean behavior and inter-individual differences characterized by some hyperparameters $\vartheta$ (Fig. 1B). To obtain simulated results we sample groups of simulated birds from these populations and let them participate in simulated experiments. For example, in one experiment[9] the caching behavior of $N_{\mathrm{experiment}} = 8$ birds was studied for two experimental conditions. We created $10^5$ groups, each containing $N = N_{\mathrm{experiment}} = 8$ simulated birds and analyzed their behavior with the same criteria as in the experiment (Figs. 1D and 2A and "Methods"). Like real birds, simulated birds perceive items of different kinds of food and potential cache sites and they perform different actions, like eating food items, caching food items or inspecting a cache site for available food. For each model and experiment, the hyperparameters $\vartheta$ are adjusted by likelihood-free inference (Fig. 1B, "Methods"). We also fitted each model to all experiments jointly, such that the model-specific hyperparameters $\vartheta$ are the same for all experiments, but the simple models we considered fail to capture the strong inter-experimental variability, which is potentially due to seasonal or other unobserved effects (Appendix). To investigate reproducibility, we also ran simulations with 10 times more subjects than in the real experiments (Fig. 2B). We do not discretize time but respect in all

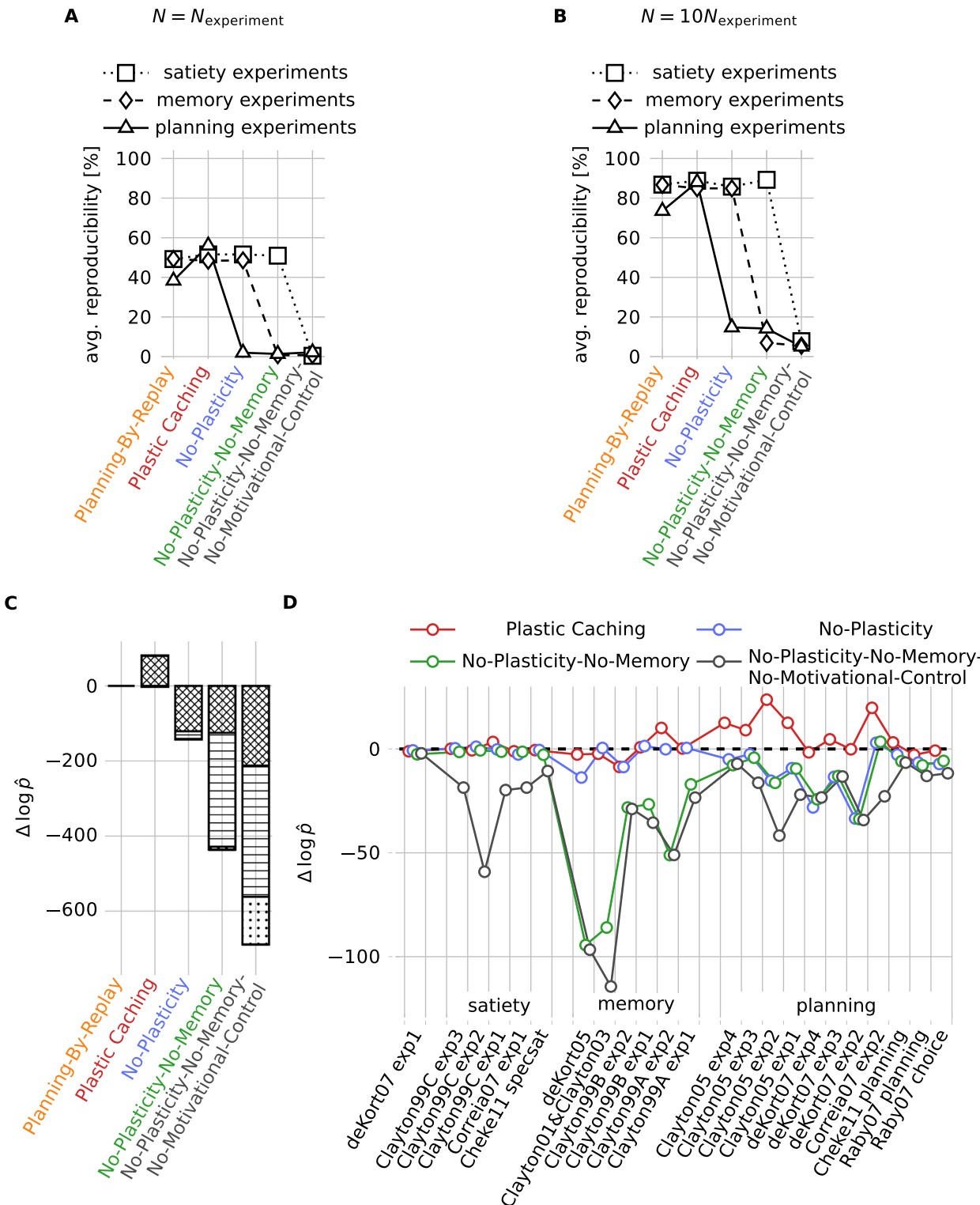

**Fig. 2 | Fitting results and model comparison. A** On average, 48–55% of experiments simulated with the Plastic Caching Model reproduce the significance level of the most relevant statistical tests, if the same number of subjects is used in the simulations as in the experiments (Methods). Without plasticity (No-Plasticity Model) performance decreases only for the planning experiments, lesioning additionally the associative memory (No-Plasticity-No-Memory Model) and motivational control (No-Plasticity-No-Memory-No-Motivational-Control Model) additionally decreases performance in memory and satiety experiments. The control model Planning-By-Replay is not better than the Plastic Caching Model. **B** If 10 times more subjects are used in the simulations than in the experiments, 85-90% of the most significant statistical tests can be reproduced. Assuming a 10% false discovery rate in the experiments, this means that the Plastic Caching Model can reproduce almost all experimentally observed effects. **C** Models with lesions perform worse than the Plastic Caching Model and the Planning-By-Replay Model as shown by the approximate log-likelihood $\Delta \log \hat{p}$ summed across all experiments and measured relative to the Planning-By-Replay Model (hatched: planning experiments, lines: memory experiments, dotted: satiety experiments). **D** Performance $\Delta \log \hat{p}$ relative to Planning-By-Replay Model for all 28 individual experiments. The 6 experiments of Clayton01& Clayton03 build upon each other and are therefore shown as a single point. DeKort07 exp1 is a control experiment.

experimental protocols the actual durations. We do not explicitly simulate movements of birds within their cage or the actual visual input; instead we assume that sensory processing in a bird's brain enables its memory and decision system to access a neuronal representation of all relevant objects.

Our main model, the Plastic Caching Model, contains modules of motivational control (green in Fig. 1C), associative memory (blue in Fig. 1C) and a caching module that relies on known synaptic plasticity rules (red in Fig. 1C, "Methods"). To study the relative importance of these modules, one or several modules can be lesioned. We also compare the Plastic Caching Model to a Planning-By-Replay Model ("Methods"), where the caching module is replaced by an explicit planning module.

### Actions depend on the available objects and the birds' motivational state

We model action selection as a random process with probabilities that depend on experiences of the simulated birds. We assume an attention mechanism that allows a bird to attend to a randomly selected object in the cage. The bird performs an action on that object if the preference is higher than a random threshold; otherwise it attends to another object until some action is selected ("Methods").

The motivational control module makes the eating and caching probabilities depend on hunger variables that are influenced by the recent eating history. We model hunger with several variables $h_f$ to capture specific satiety. Eating one type of food with a high fat concentration may decrease the first hunger variable more than the second one, whereas another type of food with a high protein concentration decreases the second variable more than the first one. If we focus only on the five experiments that measure effects of specific satiety, the motivational control module is necessary and sufficient to assure good performance; lesioning the plastic caching module (No-Plasticity Model, i.e. fixed caching probabilities) or the associative memory module (No-Plasticity-No-Memory Model, i.e. fixed caching and inspection probabilities) does not impair performance, whereas lesioning also the motivational control leads to a strong drop in performance (No-Plasticity-No-Memory-No-Motivational-Control Model, i.e. fixed caching, inspection and eating probabilities. See Fig. 2).

Whereas the motivation for eating depends undeniably on satiety, the caching behavior may be independent of hunger, as jays are known to cache predominantly in seasons when food is abundant[2]. Therefore, we compared the motivational control module to alternative motivational control models where caching is independent of the recent caching and eating history (unmodulated caching) or where caching depends only on the recent caching history (caching-modulated caching; "Methods"). We found that models without any motivational control reproduce the five specific satiety experiments clearly worse than models with hunger-modulated caching, whereas models with caching-modulated caching perform better than hunger-modulated caching models on one experiment (Clayton99C exp1) and worse on two experiments (Clayton99C exp3 and Cheke11 specsat; see Fig. S8).

### Remembering the what-where-when of caching events with an associative memory

If we extend the tests to also include the 11 memory experiments, we cannot lesion the associative memory module without a large loss in performance (Fig. 2). The module of associative memory stores caching events in a hetero-associative neural network[26] with memory consolidation at the systems level[27]. In the Plastic Caching Model (Methods), the features of the cache site ('where') get associated with the cached food ('what') by growing synaptic weights $w_{fx}^{(1)}$ (blue in Fig. 1C) between cache-site-feature neurons $\phi_x$ and first-layer food-type memory neurons $\phi_f^{(1)}$. Synaptic growth is governed by a neo-Hebbian three-factor learning rule[28]. In these rules the co-activation of

a pre- and postsynaptic neuron tags a synapse as eligible for change, but strength and sign of the actual synaptic change depends on a modulatory signal which arrives at the synapse up to a few seconds later. The simultaneous focus of a simulated bird on a caching tray $A$ and a food item $f$ activates pre- and postsynaptic neurons of memory synapses, leading to Hebbian eligibility traces. These traces are turned into actual synaptic growth, if a subsequent modulatory signal communicates that the simulated bird has indeed cached food item $f$ in caching tray $A$. Some consolidation mechanism (e.g. ref. 27) moves memories over time from synapses targeting one layer of memory neurons $\phi_f^{(l)}$ to synapses targeting the next layer of memory neurons $\phi_f^{(l+1)}$, which allows a flexible readout of how long ago a caching event happened ('when'). At retrieval time, the perception of a cache site triggers recall of the associated food type by activating memory neurons in the layer that corresponds to the time passed since caching (Fig. 1C). Synaptic readout weights are adaptive and allow the simulated birds to adjust the preferences for inspecting different cache sites. When the simulated birds experience, for example, degraded worms that were cached for a certain duration $T$, these readout weights decrease via a neoHebbian reinforcement learning rule[28], where the modulatory third factor signals the unpleasant encounter with the degraded worm. This lowers the future preference for inspecting cache sites containing worms of age $T$ ("Methods").

The second experiment reported in ref. 6 (included in Clayton01&Clayton03 in Fig. 2D) is one of the most demanding memory retrieval experiment with birds. In this experiment the birds cached on days 1, 2 and 3 in caching trays 1, 2 and 3, respectively. They were allowed to cache peanuts in one half of each caching tray and crickets in the other half of each caching tray. On day four, three days after caching in tray 1, the birds recovered their caches from tray 1. On day five they recovered their caches from tray 2. Birds in the group with manipulated trays found fresh peanuts and decayed crickets. These two recovery trials informed them that crickets decayed after 3 days, which was contrary to earlier experiences where they found fresh crickets after 3 days. On day six, they were allowed to inspect tray 3. As one would expect from birds that can generalize their experience from days four and five, they searched less for crickets than for peanuts.

In all models with associative memory (Plastic Caching Model, No-Plasticity Model, Planning-By-Replay Model), the caching events lead to the formation of synaptic contacts between caching-tray neurons $\phi_x$ and memory neurons $\phi_f^{(1)}$ (blue in Fig. 3A). Because different trays are used, different caching tray neurons $\phi_x$ are active on each of the first three days (c.f. Fig. 3A, B). However, the same memory neurons $\phi_f^{(1)}$ for $f = $ Peanut and $f = $ Cricket get activated on all three days. Over several nights, memory consolidation moves the newly formed synaptic contacts to post-synaptic neurons $\phi_f^{(2)}$, $\phi_f^{(3)}$ and $\phi_f^{(4)}$. On day four, the perception of caching tray 1 activates the corresponding caching-tray neurons, which activates $\phi_f^{(4)}$ through the synaptic contacts formed by memory consolidation. The food type $f$ of the activated memory neuron depends on which half of the caching tray the bird is looking at. The feedback after observing a decayed cricket $f = $ Cricket lowers the synaptic retrieval weight $v_{\text{Cricket}}^{(4)}$ (Fig. 3C; Eq. (5), "Methods"). This synaptic weight is further lowered on day five, where again decayed crickets are experienced. Consequently, on day six, when $\phi_{\text{Cricket}}^{(4)}$ is again activated while focusing on one half of tray 3, the retrieval weight is so low that there is only a small probability of inspecting the part of the caching tray containing crickets.

Because the weight update involves only the inputs and outputs of associative memory (blue in Fig. 3C), but not the plasticity of the caching weights (red in Fig. 3C), no differences are expected between the Plastic Caching Model, Planning-By-Replay Model and No-Plasticity Model whereas a lower performance is expected and observed for No-Plasticity-No-Memory Model and No-Plasticity-No-Memory-No-Motivational-Control Model (c.f. Fig. 2D, entry Clayton01&Clayton03).

Day 1: Caching in Tray 1

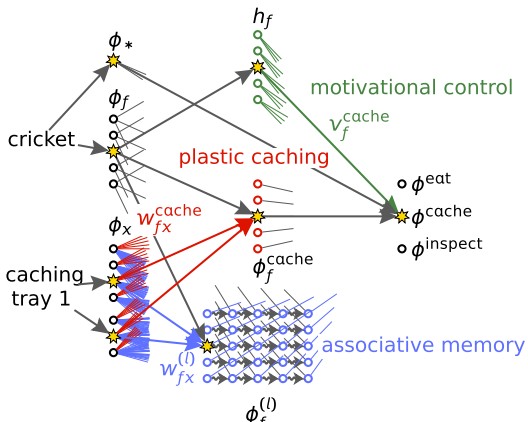

Day 3: Caching in Tray 3

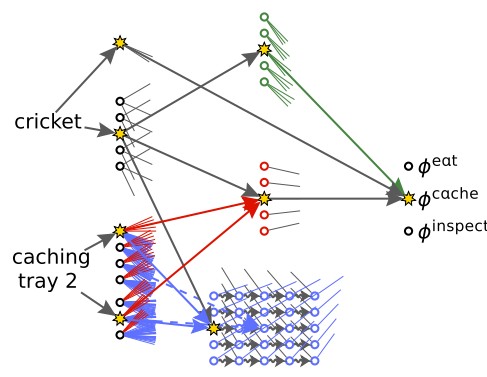

Day 4: Retrieval in Tray 1

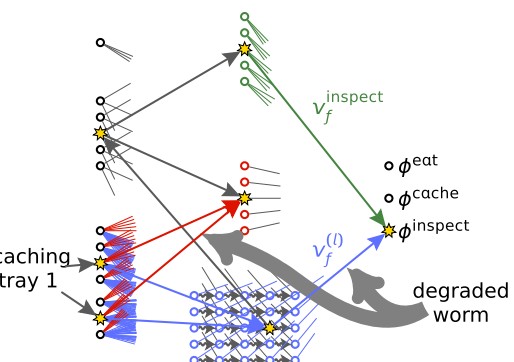

Day 6: Retrieval in Tray 3

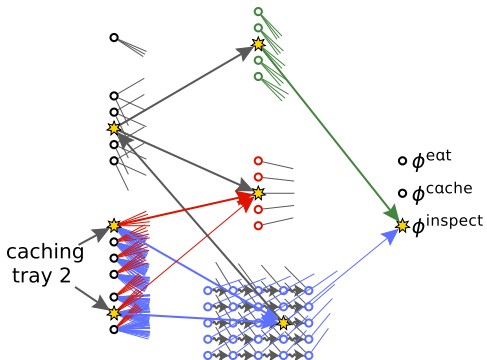

**Fig. 3 | Flexible cache memories.** In the second experiment of Clayton03, birds cached crickets on **day 1** in caching tray 1. In the Plastic Caching Model, an available food item activates a food-type neuron $\phi_f$ and the food indicator neuron $\phi_*$ and a caching tray activates cache-site neurons $\phi_x$ that code for the tray's appearance and position in the cage. The caching preference $\phi^{\text{cache}}$ depends on the current motivational state $h_f$ weighted by $v_f^{\text{cache}}$ and the caching weights $w_{fx}^{\text{cache}}$. Memory weights $w_{fx}^{(1)}$ (blue arrows) are tagged when the bird evaluates the caching preference and grown after successful caching. Caching on day 2 in tray 2 (not shown) and on **day 3** in caching tray 3 (same position, different appearance than tray 1) leads to the formation of new memory weights (blue arrows). Memory consolidation over night (wiggly gray arrows) transferred the memory of the caching event on day 1 to weights that target the third layer of the associative memory (dashed blue arrows).

The caching event on day 2 is stored in memory weights that target the second layer (not shown). Perceiving caching tray 1 on **day 4** activates the memory of the food type cached in this tray. This activates the foodtype neuron $\phi_f$, the foodtype-specific caching neuron $\phi_f^{\text{cache}}$ and the readout of the motivational state. The food indicator neuron $\phi_*$ is inactive, because the food item is not actually perceived but only remembered. The inspection preference $\phi^{\text{inspect}}$ depends on the current motivational state $h_f$ weighted by $v_f^{\text{inspect}}$ and the memory readout weights $v_f^{(4)}$. The unpleasant feedback of the degraded recovered cricket (thick gray arrows) decreases the caching weights $w_{fx}^{\text{cache}}$ and the memory readout weights $v_f^{(4)}$. These weights are further lowered on day 5 (not shown). Consequently, on **day 6** the bird has a lower preference of inspecting the tray where it cached a cricket three days ago.

## Model-free reinforcement learning is sufficient to explain the adaptation to anticipated future needs

The 11 planning experiments require additionally to motivational control and associative memory a mechanism to adapt the caching strategy. This adaptation is implemented in the Plastic Caching Model with synaptic strengths $w^{\text{cache}}$ (red in Fig. 4A) that undergo plasticity following two simple principles. First, the weights associated with available caching sites slowly increase for all food types a simulated bird feels hungry for, which explains findings in experiments without retrieval trials during training[10]. Second, the retrieval attempts of caches cause synapses $w^{\text{cache}}$ (Fig. 4B) to decrease for birds that found degraded food or were unsuccessful in finding cached food and to increase for successful birds via a neoHebbian plasticity rule[28]. Because associative recall of memories reactivates the relevant pre- and post-synaptic neurons, such a rule enables delayed reinforcement learning of those weights that determined caching decisions potentially long ago (Fig. 4A). We compare the Plastic Caching Model to a formalization of the mental-time-travel hypothesis: the Planning-By-Replay Model. This model adapts the caching strategy with an explicit planning

module that records the sequence of available caching trays, successful and unsuccessful retrieval attempts and the hunger levels that were perceived when these trays were available (Fig. 4C). Before caching a food item in a certain tray, those positions in memory are searched that match best the current context. Starting at these positions, the sequence of events is replayed for a few steps and the preference of caching is influenced by the outcome of the replayed retrieval attempts ("Methods"). Despite similarities with hippocampal replay[17,29], which would be consistent with the replay-and-plan module and models of mental-time-travel in general, we have not yet found a simple implementation of the replay-and-plan module (orange box in Fig. 4C) in terms of neural network dynamics and plasticity rules. In fact, a precise hypothesis of neural implementations of mental-time-travel requires much more than hippocampal replay, as it would have to specify which aspects of the detailed multi-sensory processing stream are stored in the hippocampal replay memory, how the memory system can efficiently be queried, and how the outcome of multiple replayed episodes are combined to reach a decision for the next action ("Methods").

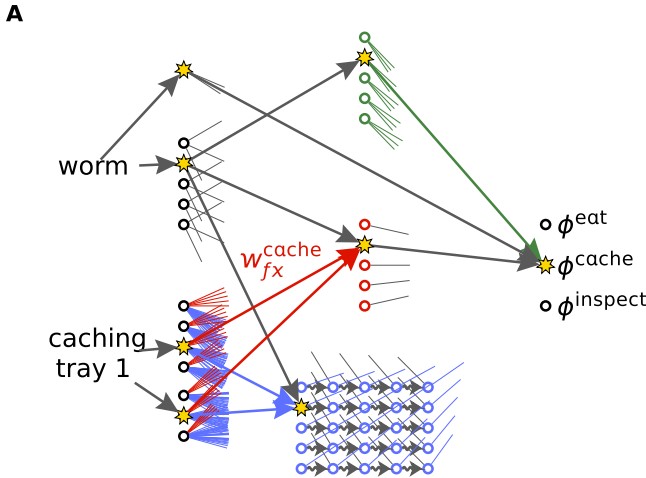

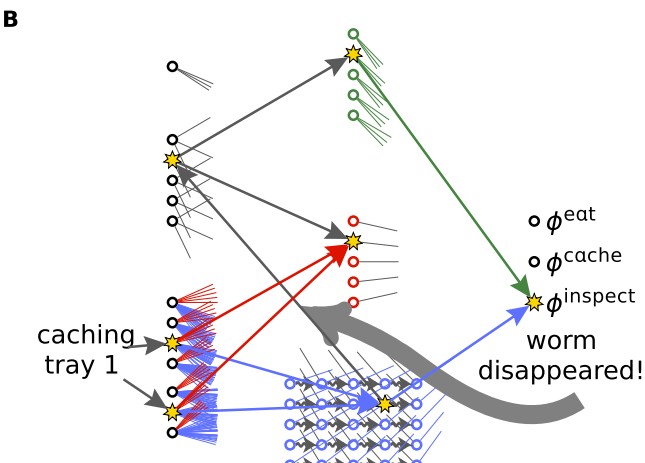

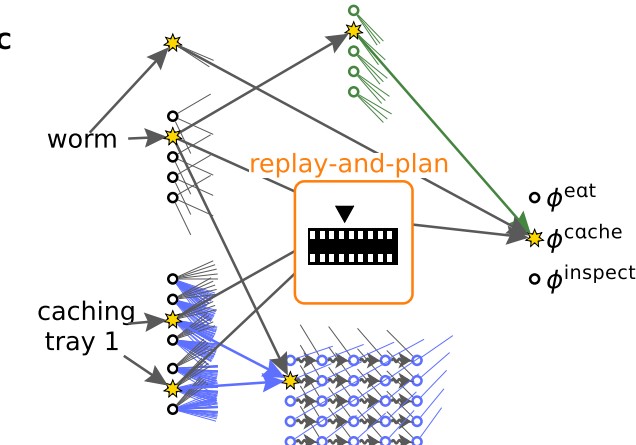

**Fig. 4 | Adaptation to anticipated future needs. A** The preference of caching worms in a given tray depends on the red caching weights $w_{fx}^{cache}$. **B** One day later, when the bird inspects the caching tray, the memory of the cached food is reactivated and, in particular, the pre- and postsynaptic neurons of the caching weights. After discovering that the cached food item has disappeared, a feedback signal acts as a third factor in a neoHebbian plasticity rule, thereby decreasing the caching weight. **C** The Planning-By-Replay Model memorizes the sequence of events (indicated with film strip). The caching preference is determined by searching for positions in memory that match best the current context (indicated with black triangle) and evaluating the preference of a caching action under the assumption that the sequence of future events resembles the sequence of events following the matched positions. All other components of the Planning-By-Replay Model are the same as in the Plastic Caching Model.

In the fourth experiment in ref. 9, birds were allowed to cache worms in tray A, whereas they could not cache in Perspex-covered tray B. One day later the Perspex covers were removed and the birds were allowed to inspect all trays. For birds in the pilfered group (blue in Fig. 1D) the experimenter removed the caches from tray A and placed them in tray B, whereas the caches of the control group were not manipulated. After two such training trials, when birds had on a later day the choice to cache either in tray A or B (not covered), birds in the pilfered group preferentially cached in tray B, whereas birds in the control group cached preferentially in tray A.

In the Plastic Caching Model, the unsuccessful retrieval attempts of the caches in tray A cause a decrease of the caching weight (Fig. 4B) of tray A for the birds in the pilfered group, whereas for the control group the successful retrievals from tray A lead to a strengthening of this caching weight (Eqs. (11)–(14), "Methods").

In the Planning-By-Replay Model, the retrieval trials are stored in the replay memory together with the information of whether the trial was successful or not. When the birds decide where to cache, previous caching events with context similar to the current one are accessed in the replay memory. From these accessed caching events, the replay memory is played forward until retrieval events from the same caching tray are encountered in memory. For birds that encounter unsuccessful retrieval events in memory the probability of caching is lower than for birds with successful retrieval events.

Because the above explanations of the experiments require plasticity of the caching weights or explicit planning, a model with lesions performs worse than the Plastic Caching Model or the Planning-By-Replay Model (cf. Fig. 2D, entry deKort07 exp4).

## Discussion

The Plastic Caching Model is sufficient to reproduce the experimental findings in 28 behavioral experiments with food-caching corvids. It is a memory-augmented reinforcement learning model in continuous time, where action selection and reward signals in simulated birds depend on external stimuli, motivational states and associative memories of past caching events. Plastic retrieval and caching weights enable adaptation to anticipated future needs.

The individual modules of the Plastic Caching Model are deliberately kept simple; the available data does not sufficiently constrain more sophisticated models. For example, little is known about how hunger sensitive neurons modulate behavior[30]. Interestingly, however, the mere visual perception of a food item transiently reduces the activity of hunger sensitive neurons in the hypothalamus of mice[31], which could potentially influence the action preferences similarly to how the motivational state modulates the preferences in our models. Further experiments on how motivational states influence behavior are needed before more refined models can be formulated.

The associative memory module with systems memory consolidation represents the time since a caching event with 'chronological organization'[32]: the identity of the memory layer that contains an active neuron at the moment of retrieval indicates the age of a memory. The specific implementation with moving synaptic connections illustrates the essential idea of how the age of memories can be retrieved in an associative memory with systems memory consolidation, but other implementations are possible (ref. 33, Methods). Alternative associative learning models like 'associative chaining', 'time tagging' or 'strength coding' would require a sophisticated neural mechanism to determine the age of memories[33] and to account for the

generalization observed in Clayton03 exp2[6]. Although there is evidence for sharp-wave ripples during sleep in the hippocampus of the food-caching bird species tufted titmice[34] and sharp-wave ripples are believed to be important for systems memory consolidation[35], future experiments are needed to determine if the time information is memorized with systems memory consolidation and an associative learning mechanism and, if so, which one of several possible associative learning mechanisms[33] is actually implemented in the brains of food-caching birds.

The multi-step consolidation mechanism in the associative memory module leads to sparse and non-overlapping activity patterns of recalled memories, which enables flexible learning of food-specific degradation[4–6] and ripening intervals[7]. This model is also consistent with experiments where magpies were trained to retrieve objects of one color for one retention interval and objects of another color for another retention interval[36]. The current model is flexible enough to learn, for example, that food of a certain type is palatable after retention intervals of 2 and 4 days and degraded after 1 and 3 days. A modified multi-step memory consolidation process may lead to more distributed and less flexible neuronal representations of retention intervals. To discriminate between different consolidation processes and neural representations of the age of a memory, future experiments should probe the limits of learnability and generalization to untrained retention intervals.

The associative memory module in the Plastic Caching Model encodes what kind of food was cached where and how long ago. It can thus be seen as a model of episodic-like memory[37], also called what-where-when memory[38]. It is, however, not a model of an episodic memory systems that enables autonoetic consciousness or mental time travel[39]. In the simulated birds, the associative memory module serves the sole purpose of coping with partial observability; it does not participate in constructive planning processes[17,22].

What looks like planning in the experiments is achieved in the Plastic Caching Model with model-free reinforcement learning, implemented with synaptic changes at the moment of retrieval. These synaptic changes follow neoHebbian three-factor learning rules which are supported by experiments in several brain areas[28,40,41]. In the birds' brains the plastic caching module (red in Fig. 1C) could be implemented by the same anatomical structure as the associative memory module (blue in Fig. 1C), because both modules receive inputs from the same neurons, they influence the action preferences and their synapses undergo neoHebbian plasticity. There is, however, a key difference between these two modules: the associative memory module keeps track of time whereas the plastic caching module does not. Our implementation of a what-where-when memory in the associative memory module allows simulated birds to learn, for example, that unripe berries cached at a warm place are palatable after a few days, or that little pieces of meat are better preserved at cold and dry places than at warm and humid places, if the warmth and humidity of a cache site are part of the perceived cache-site features. For the plastic caching module, time is irrelevant, because the birds only need to learn, for example, that little pieces of meat should preferably be cached at cold and dry places or pilfered sites should be avoided.

Computational modeling is increasingly recognized as a complementary tool to statistical hypothesis testing in experimental neuroscience and psychology[42]. Although we used the reported p-values in our likelihood-free model fitting procedure ("Methods"), we treated them as summaries of the raw data without special meaning, similarly to other summaries of the data like means and standard deviations. Fitting computational models would be simplified with more detailed raw data than usually collected in these behavioral experiments: knowing for each bird the point in time when each action was taken would enable likelihood-based model fitting ("Methods").

Our approach of formalizing experimental protocols in a domain-specific and model-independent language is an example of how

**Table 1 | Properties of the three types of objects** `Food`, `Tray` **and** `InspectionObserver`

| Food | Tray | InspectionObserver |
|---|---|---|
| freshness $\in [0, 1]$ | appearance $\in \mathbb{N}$ | tray_appearances $\in$ lists($\mathbb{N}$) |
| eatable $\in$ {true, false} | position $\in \mathbb{N}$ | |
| cacheable $\in$ {true, false} | open $\in$ {true, false} | |
| amount $\in \mathbb{N}$ | eatable_items $\in$ lists(`Food`) | |
| type $\in$ {mealworm, waxworm, peanut, suet_pellet, pinenut, kibble, cricket, pineapple, salami, stone, maintenance_diet} | | |

The set lists(X) denotes all possible lists of arbitrary length with elements in X. In the actual experiments, caching trays are usually made visually distinct by attaching them to a unique structure of Lego Duplo. We model this feature with different appearance values. Sometimes the caching trays where made inaccessible by covering them with Perplex. In this case we set open = false.

experimentalists and modelers could communicate to foster computational modeling. Experimentalists who publish raw data together with experimental protocols in a formal domain-specific language enable modelers to test existing models on new evidence. Existing models help to design new experiments and estimate the number of subjects needed to reach a desired statistical power, because the models allow to scrutinize novel experimental protocols in simulations prior to actually performing the experiment. As an example we propose a behavioral experiment, where we expect a moderate number of subjects to be sufficient to discriminate between the Plastic Caching Model and the Planning-By-Replay Model (Figs. S6 and S7). With an extension of the domain-specific language to formalize future experiments that measure physiological quantities and behavior simultaneously, the Plastic Caching Model could also be probed at the neuronal level.

Although the Plastic Caching Model has all the features to reproduce the experimentally observed behavior, some simulated repetitions of the experiments fail to reach significance on the key statistical tests with the low number of subjects typically used in the experiments (Fig. 2A, B). This suggests an alternative explanation for the recent failure of reproducing the breakfast planning experiments with Canada jays[43]: the sample number $N_{experiment} = 6$ in this experiment may have simply been too small.

With the available 28 experiments, the Plastic Caching Model reaches at least the same performance as the Planning-By-Replay Model, indicating that planning in the form of mental time-travel is not necessary to explain the experiments. Even though the Planning-By-Replay Model is just one model of mental time-travel and other mental time-travel models are conceivable, the important point is that the Plastic Caching Model can reproduce the outcomes of the planning experiments without an explicit mental time-travel strategy. The simulated birds in the Plastic Caching Model are memory-augmented stimulus-response machines that do not perform any off-line planning[17] or imagination of what it would feel like to be in an alternative situation to the currently perceived one. We conclude that memory-augmented model-free reinforcement learning methods implemented with traditional concepts of computational neuroscience such as neoHebbian learning rules[28,40,41] augmented by multi-step memory consolidation[27] are sufficient to explain the existing large body of experiments on food-caching behavior in birds.

## Methods

All considered experiments study the caching and cache recovery behavior of jays. A typical experiment repeats the following steps for $N$ individual birds, where $N$ is between 4 and 24. (1) A few hours of food

**Table 2 | Actions for formalizing the experimental protocols**

| Action | Object | Description |
|---|---|---|
| add | `Food, Tray, InspectionObserver` | Adds an object to the simulated bird cage. |
| remove | `Food, Tray, InspectionObserver` | Removes an object to the simulated bird cage. |
| cover | `Tray` | Covers a tray, prohibiting caching and inspection from this tray. |
| uncover | `Tray` | Uncovers a tray. |
| degrade | `Tray` | Degrades all food items in a caching tray. |
| pilfer | `Tray` | Removes all food items from a caching tray. |
| count_cached_items | `Tray` | Count food items in a caching tray. |
| count_food_items | | Count food items in a simulated bird cage. |
| count_inspections | `InspectionObserver` | Count the numbers of inspections to the different trays as registered by the inspection observer. |
| first_inspections | `InspectionObserver` | Determine which tray was inspected first as registered by the inspection observer. |
| wait | | Wait for a specified number of minutes. |

deprivation raise the motivation of the bird. (2) The experimenter adds some food items and visually distinct caching trays at specific positions in the bird's cage and waits for some time, e.g. 15 minutes. (3) The caching trays and remaining food items are removed and counted. (4) While trays are outside the cage, the experimenter may or may not remove (pilfer) or degrade the food items cached in some of the trays. (5) After some waiting interval, the caching trays are returned to the cage and the number of times the bird inspects the returned trays are counted during a recovery interval. (6) After another waiting period, steps 2–5 are repeated with some variations.

**Domain specific language for experimental protocols**
The domain specific language to describe all experimental protocols consists of three types of objects, `Food`, `Tray` and `InspectionObserver` with different properties (see Table 1) and eleven actions (see Table 2). With this language and some standard control commands of programming languages, like assignments, loops and if-else-clauses, all experimental protocols can be written formally as a function that takes models as input and returns result tables (see Fig. S5).

**Model description**
**Input and output.** The simulated birds can perceive food items, caching trays and immediate outcomes of their actions. We do not simulate explicit visual scenes. Instead, we use a one-hot encoding of food items, i.e. the activity $\phi_f$ of food-type neuron $f$ is equal to one, if a food item of type $f$ is perceived or remembered; otherwise, $\phi_f = 0$ (see Table 1 for possible food types $f$). To distinguish perception from memory retrieval, there is a food indicator neuron with activity $\phi_* = 1$, if a food item of any type is actually perceived. For caching trays $x$ we use a two-hot encoding: neurons $x_a$ and $x_p$ have activity levels $\phi_{x_a} = \phi_{x_p} = 1$, if the caching tray with appearance $x_a$ is placed at position $x_p$. When there is no ambiguity, we will use the symbol $x$ to denote caching trays and indices of caching-tray-feature neurons, e.g. the activity of caching-tray-feature neurons is $\phi_x = \delta_{x,x_p} + \delta_{x,x_a}$ when caching tray $x$ is present.

Simulated birds take actions. The action set

$$\mathcal{A}^{\text{bird}} = \{\text{eat}_f, \text{cache}_{fx}, \text{retrieve}_x, \text{other}\} \qquad (1)$$

consists of eating an item of food-type $f$ (eat$_f$), caching an item of food-type $f$ in caching tray $x$ (cache$_{fx}$), inspecting and trying to retrieve food from caching tray $x$ (retrieve$_x$) or doing something unrelated to the experiment (other).

There is no immediate outcome for all actions except retrieve$_x$. If a simulated bird tries to retrieve a food item from a caching tray, the food indicator neuron is active ($\phi_* = 1$) only if a food item could be retrieved from the caching tray. Furthermore, a freshness neuron signals the palatability of a retrieved food item, i.e. $\phi^{\text{fresh}} = 1$, if the food

item is fresh and $\phi^{\text{fresh}} = 0$, otherwise. The immediate outcome signals are used to modulate synaptic plasticity (c.f. Associative memory with systems consolidation and Plasticity of the caching policy in the Plastic Caching Model).

**Table 3 | Variables**

| Symbol | Description | Minimum | Maximum |
|---|---|---|---|
| $s_f(t)$ | stomach | 0 | $\infty$ |
| $h_f(t)$ | hunger or hunger | 0 | 1 |
| $M(t)$ | associative memory | | |
| $w_{fx}^{\text{cache}}(t)$ | caching weights | 0 | 1 |

$f \in FoodTypes, x \in CachingTrays.$

**Variables and parameters.** The state of each simulated bird is charaterized by some variables that change over time (Table 3) and some constant parameters (Table 4). Variable are the neural activities $\phi(t)$, the motivational control states $s_f(t)$ (stomach) and $h_f(t)$ (hunger), synaptic weights of an associative memory system $M(t)$, and the caching weights $w_{fx}^{\text{cache}}(t)$, that connect caching-tray-feature neurons $x$ to food-type neurons $f$ (Table 3). Constant parameters (Table 4) affect the dynamics of the variables as described in Motivational control, Associative memory with systems consolidation, Plasticity of the caching policy in the Plastic Caching Model and Decision making. For each individual simulated bird the fixed parameters are sampled from a distribution adjusted to the data (c.f. Model Comparison and Fitting).

**Event-based integration.** We perform event-based integration of the variables. At each event, the variables of the system can jump to new values, whereas they change smoothly between events. Events $e_k = (t_k, a_k, o_k)$ are given by the event time $t_k \in \mathbb{R}$, the event type $a_k \in \mathcal{A}^{\text{bird}} \cup \mathcal{A}^{\text{experimenter}}$, and the immediate outcome $o_k \in \mathcal{O}$, where the birds' action set is given in Eq. (1) and the experimenters' action set

$$\mathcal{A}^{\text{experimenter}} = \{\text{add}_f, \text{add}_x, \text{remove}_f, \text{remove}_x\},$$

consists of addition and removal of food items and caching trays. Note that the domain specific language to describe the experiments contains more actions, like degrade, pilfer, count_items, that are performed out of sight of the birds and do not affect their state variables directly. There is a small set of immediate outcomes $\mathcal{O} = \{\text{pilfered}, \text{fresh\_food\_item}, \text{degraded\_food\_item}\}$ that influence synaptic plasticity through the food indicator neuron and the freshness neuron, i.e. $\phi_*(t_k^+) = 1$ if and only if $o_k \neq \text{pilfered}$ and $\phi^{\text{fresh}}(t_k^+) = 1$ if and only if $o_k = \text{fresh\_food\_item}$, where $t_k^+$ indicates the time point after $t_k$ at which the outcome is known. We do not simulate any events when

**Table 4 | Parameters**

| Symbol | Description | Minimum | Maximum |
|---|---|---|---|
| $\tau_s$ | stomach time constant | 0.5 min | 10.0 min |
| $\tau_d$ | digestion time constant | 0.0 min | 20.0 min |
| $\tau_h$ | appetite increase time constant | 50.0 min | 300.0 min |
| $n_f$ | nutritional value | 0.1 | 1.0 |
| $v_f^{eat}$ | eating preferences | 0.1 | 1.0 |
| $v_f^{cache}$ | caching preferences | 0.0 | 1.0 |
| $\eta^{eat}$ | eating bias | -1.0 | 1.0 |
| $\eta^{inspect}$ | inspection bias | -2.0 | 0.5 |
| $s^{inspect}$ | inspection pre-ference slope | 0.0 | 5.0 |
| $\eta^{cache}$ | caching bias | -1.0 | 1.0 |
| $\alpha^{fresh}$ | freshness learning rate | 0.0 | 1.0 |
| $\alpha^{reward}$ | retrieval reward learn-ing rate | 0.0 | 0.9 |
| $\alpha^{pilfer}$ | pilfer learning rate | 0.0 | 0.2 |
| $\alpha^{degrade}$ | degradation learning rate | 0.0 | 0.2 |
| $\tau^{hungry}$ | hunger–caching time constant | 100.0 min | 500.0 min |
| $w_0^{cache}$ | initial caching weights | 0.0 | 1.0 |
| $\delta^{eat}$ | eating time-out | 10.0 s | 200.0 s |
| $\delta^{inspect}$ | inspection time-out | 10.0 s | 200.0 s |
| $\delta^{cache}$ | caching time-out | 10.0 s | 200.0 s |
| $\delta^{other}$ | other action time-out | 10.0 s | 200.0 s |
| $p^{other}$ | other action preference | 0.0 | 1.0 |

The ranges [minimum, maximum] are manually set. The nutritional values, eating preferences and caching preferences are estimated for each food type $f \in$ {mealworm, waxworm, peanut, suet_pellet, pinenut, kibble, cricket, pineapple, salami}; for stones only the caching preference is estimated.

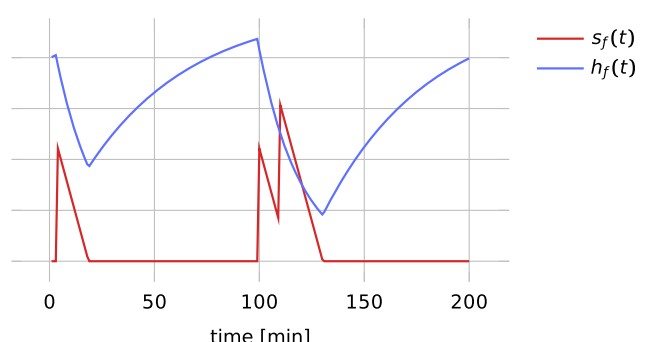

**Fig. 5 | Example trace of stomach $s_f$ and hunger $h_f$ variables.** At minutes 4, 100 and 110 the simulated bird eats an item of food type $f$, thereby increasing the stomach variable and shortly thereafter decreasing hunger.

a bird has no food items and caching trays available that are relevant to the experiment.

**Motivational control.** We model motivational control with two variables per food-type: $s_f(t)$ indicates how much food of type $f$ is in the stomach and $h_f(t)$ indicates how hungry the bird is for food of type $f$ (Fig. 5). Instead of one stomach and hunger variable per food type one could assign one variable per nutrient class, e.g. carbohydrates, fats, fiber, minerals, proteins, vitamins, and water. However, the mapping between food types and nutrient classes is non-trivial and in the experiments the food types and quantities were chosen such that the birds satiated on one food clearly

continued desiring and eating another food. Therefore we work directly with food types.

The stomach variables evolve according to

$$\frac{ds_f}{dt} = -\frac{1}{\tau_s}\left[s_f > 0\right] + \sum_k n_f \delta(t - t_k)\left[a_k = \text{eat}_f\right], \quad (2)$$

where $\tau_s$ is a stomach time constant, the Iverson bracket $[P]$ is 1 if the statement $P$ is true and 0 otherwise, $n_f$ is the nutritional value of food of type $f$, $\delta(.)$ is the Dirac delta distribution, and $t_k$, $a_k$ are event times and event types, respectively. The hunger variables are coupled to the stomach variables through

$$\frac{dh_f}{dt} = -\frac{h_f}{\tau_d}\left[s_f > 0\right] + \frac{1 - h_f}{\tau_h}\left[s_f = 0\right] \quad (3)$$

where the digestion time constant $\tau_d$ determines how fast hunger decrease while there is some food in the stomach and $\tau_h$ determines how fast it increases otherwise. Whereas the stomach variable increases immediately with every eaten food item and decreases linearly as food is digested, hunger decreases slowly during digestion, because food absorption is not immediate and major hunger satiation signals arise from the gut[30]. The value 'zero' of the stomach variable should not literally be understood as indicating a completely empty stomach; rather it is the emptiness level of the stomach at which a bird's hunger feeling starts to increase again. If maintenance diet is available to the birds, we do not explicitly simulate eating events, but integrate Eq. (3) under the assumption $s_f > 0$ for all $f$. The hunger variables $h_f(t)$ critically influence decision making (see Decision making).

In the caching-modulated caching model of motivational control we use additionally the caching motivation variables $c_f$ that evolve according to

$$\tau_d \frac{dc_f}{dt} = 1 - c_f(t) - c_f(t)c_0 \sum_k \delta(t - t_k)\left[a_k = \text{cache}_f\right] \quad (4)$$

where $c_0 \in [0, 1]$ is a fitted parameter that controls how strongly the caching motivation variables $c_f$ decreases when caching an item of type $f$. In other words, whenever a simulated bird caches an item of type $f$, the caching motivation $c_f$ for food $f$ decreases by the amount $c_f(t) \cdot c_0/\tau_d$ towards zero and in the absence of caching events it increases exponentially to one with time-constant $\tau_d$. If the fitted parameter $c_0$ is large, few caching events suffice to bring the caching motivation close to zero.

**Associative memory with systems consolidation.** Memory about previous caching events is implemented in a simple associative (content-addressable) memory system $M(t)$. The memory system $M(t)$ consists of 7 memory sub-networks and readout weights $v_f^{(l)}$. Each sub-network $l = 1, ..., 7$ contains neurons $\phi_f^{(l)}$ that are activated by actually perceived or remembered food of type $f$. Initially, all synaptic weights $w_{fx}^{(l)}$ from caching-tray-feature neurons $\phi_x$ to food-type neurons $\phi_f^{(l)}$ in layer $l$ are zero. Each caching event increases a synaptic consolidation variable $\bar{w}_{fx}^{(1)}(t)$ by 1. The effective synaptic strength of connections onto memory neurons $\phi_f^{(1)}$ on layer $l = 1$ is $w_{fx}^{(1)}(t) = H(\bar{w}_{fx}^{(1)}(t - 1\,\text{hour}))$, with Heaviside function $H(x) = 1$ if $x > 0$ and $H(x) = 0$ otherwise. Implementing a simple systems consolidation process, active weights $w_{fx}^{(l)} = 1$ and variables $\bar{w}_{fx}^{(l)}$ in networks $l = 1, 2, 3, 4$ are copied to networks $l + 1$ and erased (set to zero) in network $l$, every day, i.e. $w_{fx}^{l+1}(t + 1\,\text{day}) = w_{fx}^{(l)}(t)$ and $w_{fx}^{(1)}(t + 1\,\text{day}) = 0$ unless there was a caching event during $t$ and $t + 1\,\text{day} - 1\,\text{hour}$. Active weights stay in network 5 for 3 days before moving to network 6, where they stay for 8 days before moving to the last layer of the memory network. In the experiments the birds never cached on different days in the same caching tray. Also in the wild, caching new items at a site where some

food items are already cached is unlikely, given that these birds are scatter hoarders. In our model, repeated caching at the same site on multiple days would lead to replacement of the old memory trace, e.g. caching peanuts at $x$ on day 1 and caching again peanuts at $x$ on day 3 would lead to the deletion of the weights targeting layer 3 and the growth of new weights to layer 1. Again, we choose a simple 'one-hot code' in time, where each memory about a caching event is encoded in exactly one neuron of the memory network at a time, but more distributed representations with gradual shifts would be possible.

Recall occurs when a caching tray is perceived at time $t$. Perception of caching tray $x$ activates a cache-position and a cache-appearance neuron $\phi_{x_p}(t) = \phi_{x_a}(t) = 1$, which in turn activates those food neurons $\phi_f^{(l)}$ that have an active weight $w_{fx}^{(l)} = 1$, i.e.

$$\phi_f^{(l)}(t) = \sigma\left(\sum_x w_{fx}^{(l)} \phi_x(t)\right), \qquad (5)$$

where $\sigma(x) = \min(1, \max(0, x))$ is the activation function. Each actual inspection of a cache site $x$ reduces the variable $\bar{w}_{fx}^{(l)}$ by 1, such that after sufficiently many inspection events the weight $w_{fx}^{(l)}$ becomes zero. With this recall mechanism, the identity $l$ of the layer of the memory network that contains an active food neuron codes for the age of the memory: if the only caching event of food of type $f$ at site $x$ occurred three days ago, only $\phi_f^{(3)}$ becomes active. With the systems consolidation process described above, the corresponding age intervals of the memory networks are: layer 1: $[0, 1)$ day, 2: $[1, 2)$ days, 3: $[2, 3)$ days, 4: $[3, 4)$ days, 5: $[4, 7)$ days, 6: $[7, 15)$ days and 7: $[15, \infty)$ days. Each neuron in the memory network signals its activity through synapses of strength $v_f^{(l)}$ to a retrieval preference neuron $\phi^{\text{retrieval}}$. These weights $v_f^{(l)}$ encode the expected freshness of food of type $f$ that was cached at a time that lies in the corresponding age interval of network $l$. These weights are subject to experience-dependent plasticity: if the experienced freshness during a successful retrieval event differs from the expected value, the weight changes according to

$$\Delta v_f^{(l)} = \alpha^{\text{fresh}}(\phi^{\text{fresh}} - v_f^{(l)}), \qquad (6)$$

where $\phi^{\text{fresh}} = 1$ indicates that the retrieved food was palatable and $\phi^{\text{fresh}} = 0$ indicates the opposite.

To summarize, the state of the memory network at time $t$ is given by the variables

$$M(t) = \left\{\phi_f^{(l)}(t), \bar{w}_{fx}^{(l)}(t), w_{fx}^{(l)}(t), v_f^{(l)}(t)\right\}_{l=1,\dots,7}. \qquad (7)$$

This associative memory system with systems consolidation is just one hypothesis of automatic processes that keep track of when an event was memorized[33]. One alternative is to grow connections to all layers of the memory network at the moment of storage, while maintaining a time-dependant activity pattern at retrieval through synaptic connections that disappear at different rates, e.g. the connections to the first layer could disappear after one day, whereas those to other layers disappear later. In this case, a young memory would be characterized by many neurons being active during recall and an old memory by few neurons being active during recall. Another alternative, without multiple layers and systems consolidation, is to store the 'what-where' information together with a 'when' tag, like a time stamp, or a 'context' tag that allows to reconstruct the 'when' information. Implementing the 'when' information with a time tag or the number of active neurons during recall has computational disadvantages to the sparse code of the memory network $M(t)$, because quickly learning flexible rules based on the what-where-when of recalled events is easiest with linear readout, when the input to the linear readout is sparse, ideally, one-hot coded[33]. But further experiments are needed to discover the actual implementation of the what-where-when memory in food caching birds.

**Decision making.** At any given moment in time the simulated birds have $1 + n_f + n_f \times n_x + n_x$ actions available (1 other, $n_f$ eat, $n_f \times n_x$ cache, $n_x$ inspect), where $n_f$ and $n_x$ are the currently available numbers of food types and cache sites, respectively. Actions are selected by weighted sampling from the available actions with the weights given by the preferences of the available actions. This is implemented iteratively as follows:

1. Compute the preference $p_a$ of a randomly selected available action $a$.
2. Sample $z$ from a uniform distribution over the interval $[0, 1]$.
3. Choose action $a$ if $p_a > z$, otherwise repeat steps 1. – 3.

After every action the simulated birds wait for a random duration sampled from a uniform distribution over $[1, \delta^{a_k}]$ seconds before taking the next action.

The preference $p_{\text{other}}$ is a bird-specific preference for doing something unrelated to the experiment. The computation of all other preference $p_a$ could be implemented by a visual attention mechanism that focuses randomly on an available food item or caching tray. For example, focusing on a food item of type $f$ would activate the food-indicator neuron and one food-type neuron $\phi_* = \phi_f = 1$; all other input neurons would be inactive. The output neuron with activity

$$\phi^{\text{eat}} = \sigma\left(\sum_{f'} v_{f'}^{\text{eat}} h_{f'} \phi_{f'} + \eta^{\text{eat}} + \phi_* - 1\right) = \sigma\left(v_f^{\text{eat}} h_f(t) + \eta^{\text{eat}}\right) = p_{\text{eat}f} \quad (8)$$

would than compute the preference $p_{\text{eat}f}$ of eating this item (see Fig. 1C). In Eq. (8), $\sigma(x) = \min(1, \max(x, 0))$ is an activation function, $h_f$ are the current values of the hunger variables (c.f. Motivational control) and $v_f^{\text{eat}}$ are fitted eating preference parameters. The preferences for caching a food item of type $f$ in caching tray $x$ are computed as

$$\phi^{\text{cache}} = p_{\text{cache}fatx} = \sigma\left(w_{fx_p}^{\text{cache}} + w_{fx_a}^{\text{cache}} + v_f^{\text{cache}} h_f(t) + \eta^{\text{cache}}\right), \qquad (9)$$

where $x_p$ and $x_a$ are the position and appearance of the caching tray, respectively, and $w_{fx_p}^{\text{cache}}$, $w_{fx_a}^{\text{cache}}$ are corresponding weights that undergo plasticity in the Plastic Caching Model (c.f. Plasticity of the caching policy in the Plastic Caching Model). In the unmodulated caching model we drop the term $v_f^{\text{cache}} h_f(t)$ in the equation above and in the caching-modulated caching model we replace it by $v_f^{\text{cache}} c_f(t)$ (see also Fig. S8).

The preferences for inspection of caching tray $x$ are given by

$$\phi^{\text{inspect}} = p_{\text{inspect}x} = \max_{f,l} \sigma\left(\phi_f^{(l)} v_f^{(l)} + v_f^{\text{inspect}} h_f + \eta^{\text{inspect}}\right), \qquad (10)$$

where $v_f^{\text{inspect}} = s^{\text{inspect}} v_f^{\text{eat}}$ to reduce the number of free parameters.

**Plasticity of the caching policy in the Plastic Caching Model.** The caching policy is a simple stimulus-response mechanism that depends on caching-tray and food-type specific synaptic weights $w_{fx}^{\text{cache}}$ (red weights in Fig. 1C of the main text). These are the weights from presynaptic caching-tray feature neurons $\phi_x$ to post-synaptic caching preference neurons $\phi_f^{\text{cache}}$. Their changes are governed by a modulated Hebbian plasticity term (first term in Eq. (11)) and a pre-synaptic term (second term in Eq. (11))

$$\frac{dw_{fx}^{\text{cache}}(t)}{dt} = I(t)\left(m_1 w_{fx}^{\text{cache}} + m_2(1 - w_{fx}^{\text{cache}})\right)\phi_x(t)\phi_f^{\text{cache}}(t) \\ + H_{fx}(t)(1 - w_{fx}^{\text{cache}}(t))\phi_x(t) \qquad (11)$$

$$m_1 = -\alpha^{\text{pilfer}}\left(1 - \phi_*(t_k^+)\right) - \alpha^{\text{degrade}}\phi_*(t_k^+)\left(1 - \phi^{\text{fresh}}(t_k^+)\right) \qquad (12)$$

$$m_2 = \alpha^{\text{fresh}}p_f^{\text{eat}}\phi_*(t_k^+)\phi^{\text{fresh}}(t_k^+), \qquad (13)$$

where $H_{fx}(t) = \frac{1}{\tau^{\text{hungry}}}[h_f(t) > \theta^{\text{hungry}}]$ is active when a bird's hunger for food of type $f$ exceeds some threshold $\theta^{\text{hungry}} = 0.99$, $I(t) = \sum_k \delta(t - t_k)$ $[a_k = \text{retrieve}]$ and $p_f^{\text{eat}}$ is the current preference for eating food of type $f$ (see Decision making). The effect of the two modulation factors $m_1$ and $m_2$ is best understood by a case-by-case analysis. Indeed, the modulated Hebbian part of the plasticity rule in Eq. (13) can be written as a change of the caching weights after attempting to retrieve a food item of type $f$ at cache site $x$

$$\Delta w_{fx}^{\text{cache}} = \begin{cases} -\alpha^{\text{pilfer}} & \times w_{fx}^{\text{cache}} & \text{if } o_k = \text{pilfered} \\ -\alpha^{\text{degrade}} & \times w_{fx}^{\text{cache}} & \text{if } o_k = \text{degraded\_food\_item} \\ +\alpha^{\text{fresh}}p_f^{\text{eat}} & \times (1 - w_{fx}^{\text{cache}}) & \text{if } o_k = \text{fresh\_food\_item} \end{cases} . \qquad (14)$$

The gated presynaptic term in Eq. (13) leads to behavior consistent with the Compensatory Caching Hypothesis[44]: birds learn to compensate for a lack of food experienced at a certain place by caching more at this place, because the caching preference weights increases at this place.

**The caching policy in the Planning-By-Replay Model.** The Planning-By-Replay Model maintains a list $\mathcal{T} = ((m_1, t_1), (m_2, t_2), \ldots, (m_I, t_I))$ of $I$ memory items $m_i$ with their time-of-last-change $t_i$. The memory items are collections of tray-hunger-outcome associations

$$m_i = \left\{ \left(x_i^{(1)}, \boldsymbol{h}_i^{(1)}, o_i^{(1)}\right), \left(x_i^{(2)}, \boldsymbol{h}_i^{(2)}, o_i^{(2)}\right), \ldots, \left(x_i^{(J_i)}, \boldsymbol{h}_i^{(J_i)}, o_i^{(J_i)}\right) \right\}, \qquad (15)$$

where $x_i^{(j)}$ identifies a caching tray, $\boldsymbol{h}_i^{(j)}$ is a vector of hunger levels and $o_i^{(j)}$ is an outcome value in $\mathcal{O}' = \mathcal{O} \cup \{\text{not\_inspected}\}$. The outcome not_inspected is used when a simulated bird perceives the presence of a caching tray but never inspects its content. If a simulated bird interacts at time $t$ with a caching tray $x$ and observes outcome $o_t$ the memory gets updated in the following way. If the caching tray $x$ is identical to $x_I^j$ and $t - t_I < 1$ hour, $\boldsymbol{h}_I^{(j)}$ becomes $\frac{1}{2}\boldsymbol{h}_I^{(j)} + \frac{1}{2}\boldsymbol{h}(t)$, where $\boldsymbol{h}(t)$ is the currently perceived vector of hunger levels, $o_I^{(j)}$ becomes $o_t$ and $t_I$ becomes $t$. If the caching tray $x$ is different from all $x_I^j$ and $t - t_I < 1$, a new tray-hunger-outcome associations $(x, h(t), o_t)$ is appended to $m_I$. If the last change of the memory dates back more than one hour, i.e. $t - t_I > 1$ hour, a new memory item $m_{I+1} = (x, h(t), o_t)$ is created and appended to $\mathcal{T}$. This grouping of individual interactions simplifies the search for replay positions, which would be difficult in a naive memory list $((x_1, \boldsymbol{h}_1, o_1, t_1), (x_2, \boldsymbol{h}_2, o_2, t_2), \ldots)$ that keeps track of all individual interactions with all caching trays.

The set of replay indices $\mathcal{I}^*$ is determined by comparing the tray identifiers in the last three memory items to those at different positions in the memory list $\mathcal{T}$ and picking the closest matches, i.e.

$$\mathcal{I}^* = \arg\max_{i < I} \sum_{k=0}^{2} \left[ x_{I-k}^{(1)} = x_{i-k}^{(1)} \wedge x_{I-k}^{(2)} = x_{i-k}^{(2)} \wedge \cdots \wedge x_{I-k}^{(J_{I-k})} = x_{i-k}^{(J_{I-k})} \right], \qquad (16)$$

unless the closest match is zero, in which case we empty the set of replay indices $\mathcal{I}^* = \emptyset$. Using the last three memory items to query the memory list allows to search for patterns in memory with similar context. The similarity of the context is further quantified by a similarity weight

$$w_i = \frac{1}{3}\sum_{k=0}^{2} \frac{1}{J_{i-k}} \sum_{j=1}^{J_{i-k}} 1 - \left( \| \boldsymbol{h}_{I-k}^{(j)} - \boldsymbol{h}_{i-k}^{(j)} \|_1 + \left[ o_{I-k}^{(j)} = o_{i-k}^{(j)} \right] \right)/(\dim(\boldsymbol{h}) + 1).$$

To determine the preference of caching a food item of type $f$ in caching tray $x$, the set of replay indices $\mathcal{I}^*$ is determined and memory items are replayed. During replay from index $i^* \in \mathcal{I}^*$, memory items $m_{i^*+k}$ for $k = 1, \ldots, 6$ are searched for tray-hunger-outcome associations that contain the tray identifier $x$. The result of this search can be written as the set $Q(x, i^*) = \{(i, j) | x_i^{(j)} = x, i^* < i \leq i^* + 6\}$ of index pairs with matching tray-identifier $x$. For each index pair $(i, j)$ the weight $w_{ij}(f)$ for caching a food item of type $f$ is determined by computing

$$\begin{aligned} w_{ij}(f) = &\alpha^{\text{hunger}}h_{if}^{(j)} \\ &+ \alpha^{\text{fresh}}[h_{if}^{(j)} \leq \theta][o_i^{(j)} = \text{fresh\_food\_item}] \\ &- \alpha^{\text{degrade}}[o_i^{(j)} = \text{degraded\_food\_item}] \\ &- \alpha^{\text{pilfer}}[o_i^{(j)} = \text{pilfered}]. \end{aligned}$$

The distributions over parameters $\alpha^{\text{hunger}}, \alpha^{\text{fresh}}, \alpha^{\text{degrade}}$ and $\alpha^{\text{pilfer}}$ are fitted (see Model Comparison and Fitting). Finally, the weight $w_{fx}$ for caching is determined by a weighted sum

$$w_{fx} = \sum_{i^* \in \mathcal{I}^*} w_{i^*} \sum_{(i,j) \in Q(x, i^*)} \gamma^{i-i^*} w_{ij}(f), \qquad (17)$$

where $\gamma \in [0, 1]$ is a discount factor. The tangent hyperbolic of this weight $\tanh(w_{fx})$ (instead of $w_{fx_p}^{\text{cache}} + w_{fx_a}^{\text{cache}}$) is used to determine the preference of caching in Eq. (9).

The algorithmic description of the Planning-By-Replay Model illustrates three problems that a neural implementation of mental-time travel for planning needs to solve: first, what is stored in memory (Eq. (15)), second, how is the memory queried (Eq. (16)); and third, how are replayed episodes used for decision making (Eq. (17)). If food caching birds use indeed mental-time travel for planning, they probably do not store in memory every single observation and action, as if everything was recorded on a videotape. Instead they may store some compressed representation of past experiences, like the available food or the average hunger level in the afternoon of a given day. Fast hippocampal replay of recent experiences may contribute to storing such compressed representations in long-term memory, but we do not yet have a detailed hypothesis of the underlying neural processes. Likewise, querying the memory system would involve probably non-trivial neural processing. The sight of a peanut should not trigger the memory system to retrieve the myriad of past experiences with peanuts. Instead, the memory system should be queried with information that is relevant in the given context. Finally, if a query leads to recall of multiple past episodes, their content should probably be held in some working memory for further information processing and decision making, which is likely to involve intricate neural processing.

## Models with lesions

**No-Plasticity-No-Memory-No-Motivational-Control Model.** The simplest model is obtained by setting to zero at all times all state variables $h_f(t)$, $w_{fx}^{\text{cache}}(t)$ and $\phi_f^{(l)}(t)$ in the computation of the action probabilities Eq. (8), Eq. (9) and Eq. (10). In this case, the fitted parameters are the preference $p^{\text{other}}$, the biases $\eta^{\text{eat}}, \eta^{\text{cache}}, \eta^{\text{inspect}}$ and the time-out constants $\delta^{\text{other}}, \delta^{\text{eat}}, \delta^{\text{cache}}, \delta^{\text{inspect}}$.

**No-Plasticity-No-Memory Model.** In this simplified model the state variables $w_{fx}^{\text{cache}}(t)$ and $\phi_f^{(l)}(t)$ are kept zero at all times, but the hunger variables evolve according to the dynamics of the full model. The fitted parameters for this model include, in addition to the parameters of the simplest model, all food parameters and the stomach, digestion and hunger-increase time constants.

**No-Plasticity Model.** In this simplified model only the state variables $w_{fx}^{\text{cache}}(t)$ are kept zero at all times, but the hunger dynamics and the associative memory dynamics with systems consolidation evolve as in

the full model. The additional parameters are the inspection preference slope and the freshness learning rate.

## Average reproducibility

For each experiment we identify 1 to 3 relevant statistical tests (called 'most relevant tests' in the following) to support the major finding (see Supplementary Results). For each test we compare the simulated p-values $p_{sim}$ to the experimental p-values $p_{exp}$. If they are both below 0.05 or both above 0.05 we say the simulation reproduces the test. We say a simulated experiment *reproduces* the experimental result, if all 'most relevant tests' are simultaneously reproduced. The results in Fig. 2A are obtained by simulating each experiment 1000 times and computing the fraction of simulated experiments that reproduce the experimental results. Given the intrinsic variability of the experimental data (see Supplementary Results) we consider average reproducibilities above 50% as high values.

To check whether the plotted differences in average reproducibility are significant, we note that the plotted numbers can be interpreted as empirical rates of 1000 Bernoulli trials. Therefore, differences of 3% average reproducibility have a one-sided p-value of at most 0.031.

## Model comparison and fitting

For each experiment $\mathcal{E}$, the data $\mathcal{D}_{\mathcal{E}}$ collected in the experiments consists of numbers $x_i$, like means, standard errors of the mean or p-values for some statistical tests, i.e. $\mathcal{D}_{\mathcal{E}} = \{x_i | i \in \mathcal{I}_{\mathcal{E}} \cup \mathcal{P}_{\mathcal{E}}\}$, where $\mathcal{I}_{\mathcal{E}}$ is the index set of all quantities except p-values and $\mathcal{P}_{\mathcal{E}}$ is the index set of p-values for experiment $\mathcal{E}$. We extracted at least 5 (Raby07 planning) and at most 212 (Clayton99B exp1) observed quantities from figures and text of the respective publications (1568 observed quantities in total).

Models are characterized by their structure, i.e. the probability of actions given the parameters and past observations, and their distribution over parameters. This distribution describes a population of birds, rather than individual birds. Each model has a different number $M$ of parameters $\theta_i$ that are constrained to be in an interval $[l_i, u_i]$, $i = 1, ..., M$ (see Table 4). We parametrize probability densities over these intervals with the help of beta distributions

$$\theta_i = (u_i - l_i)z_i + l_i \quad p_Z(z_i; s_i, d_i) = \begin{cases} c(s_i, d_i)z_i^{f(s_i)}(1 - z_i)^{f(s_i + d_i)} & d_i < 0 \\ c(s_i, d_i)z_i^{f(s_i - d_i)}(1 - z_i)^{f(s_i)} & d_i \geq 0 \end{cases}$$
(18)

where $c(s_i, d_i)$ is the normalization constant of the beta distribution and $f(x) = \log(\exp(x) + 1) - 1$. We found empirically that this parametrization led to reasonably fast and robust optimization results. We write

$$p(\boldsymbol{\theta}; \boldsymbol{d}, \boldsymbol{s}) = \prod_{i=1}^{M} p_Z((\theta_i - l_i)/(u_i - l_i); s_i, d_i)$$
(19)

for the density of parameter vectors $\boldsymbol{\theta}$ given hyperparameters $\boldsymbol{d}, \boldsymbol{s}$. A simulated bird is characterized by a sample $\boldsymbol{\theta}$ from this probability distribution. The hyperparameter vectors $\boldsymbol{d}$ and $\boldsymbol{s}$ are fitted. In Fig. 1B of the main text we display the parameters $\boldsymbol{\theta}$ with a blue bird and use $\vartheta$ for the hyperparameters $\boldsymbol{d}, \boldsymbol{s}$.

For each simulated experiment $\mathcal{E}$, we sample $N_{\mathcal{E}}$ birds independently from the population distribution (Eq. (18)), where $N_{\mathcal{E}}$ is the number of birds used in the actual experiment $\mathcal{E}$. Although some birds participated in multiple experiments, we believe re-sampling is justified, because we do not explicitly model other effects that may influence behavior, like aging or the change of seasons. Excluded from re-sampling is exclusively the series of experiments described in[5,6]. We treat this series denoted as Clayton01&Clayton03 in Fig. 2C as one very

long experiment, because some experiments rely on the experience the birds made in earlier experiments of the same series.

For the given data and the models described in Model Description we have an intractable likelihood function

$$\ell(\boldsymbol{d}, \boldsymbol{s}) = \prod_{\mathcal{E}} p(\mathcal{D}_{\mathcal{E}} | \boldsymbol{d}, \boldsymbol{s}) \quad \text{with} \quad p(\mathcal{D}_{\mathcal{E}} | \boldsymbol{d}, \boldsymbol{s})$$
$$= \int p\left(\mathcal{D}_{\mathcal{E}} | \boldsymbol{\theta}^{(1)}, ..., \boldsymbol{\theta}^{(N_{\mathcal{E}})}\right) \prod_{j=1}^{N_{\mathcal{E}}} p(\boldsymbol{\theta}^{(j)}; \boldsymbol{d}, \boldsymbol{s}) d\boldsymbol{\theta}^{(j)},$$
(20)

where $\boldsymbol{\theta}^{(j)}$ denotes the parameters of bird $j$ and the product runs over all experiments $\mathcal{E}$ under consideration. The source of intractability in Eq. (20) are the conditional probability densities $p\left(\mathcal{D} | \boldsymbol{\theta}^{(1)}, ..., \boldsymbol{\theta}^{(N_{\mathcal{E}})}\right)$, which are marginal distributions over all possible action sequences giving rise to the reported summary statistics. Note that the likelihood would be tractable and a standard likelihood based fitting procedure could be used, if the action sequences of all birds that participated in the real experiments were known. For Fig. 2 of the main text the parameters are fitted to each experiment individually, except for the experiments in the Clayton01&Clayton03 series that are fitted jointly. See Supplementary Results for joint fits of all experiments or subsets thereof.

Because the dimensionality of the hyperparameters ($8 \leq \dim(\boldsymbol{d}) = \dim(\boldsymbol{s}) = M \leq 45$) and the data is rather high, we resort to approximate maximum likelihood estimation to find the parameters that characterize best the population of birds. We use $k$-nearest-neighbor density estimates[45] to approximate the logarithm of the likelihood function in Eq. (20). To do so, we repeat each experiment $\mathcal{E}$ with independent groups $k = 1, ..., K$ of $N_{\mathcal{E}}$ birds, each charaterized by parameters $\boldsymbol{\theta}^{(k,1)}, ..., \boldsymbol{\theta}^{(k,N_{\mathcal{E}})}$, where $N_{\mathcal{E}}$ is the number of birds that participate in experiment $\mathcal{E}$. From the simulations with index $k$, we compute the same means, standard errors of the mean and p-values $x_i^{(k)}$ as in the actual experiment. This results in simulated data $\mathcal{D}_{\mathcal{E}}^{(k)} = \{x_i^{(k)} | i \in \mathcal{I}_{\mathcal{E}} \cup \mathcal{P}_{\mathcal{E}}\}$ for experiment $\mathcal{E}$. For each simulated data set we compute the distance

$$\Delta_{\mathcal{E}}^{(k)} = \sqrt{\sum_{i \in \mathcal{I}_{\mathcal{E}}} \left(x_i^{(k)} - x_i\right)^2 + \sum_{j \in \mathcal{P}_{\mathcal{E}}} \left(s\left(x_j^{(k)}\right) - s\left(x_j\right)\right)^2},$$
(21)

where $x_i$ is the experimentally observed value, $\mathcal{I}_{\mathcal{E}}$ and $\mathcal{P}_{\mathcal{E}}$ are the index sets defined above and $s$ is a quantization of p-values given by

$$s(p) = \begin{cases} 1 & p < 0.001 \\ 2 & 0.001 \leq p < 0.01 \\ 3 & 0.01 \leq p < 0.05 \\ 4 & 0.05 \leq p < 0.1 \\ 5 & \text{otherwise}. \end{cases}$$
(22)

This corresponds roughly to computing differences of p-values on a log-scale, but it does not emphasize difference between highly significant p-values, e.g. $s(10^{-11}) = s(10^{-5}) = 1$, which reduces the variance and helps during fitting. Let us assume that these distances are ordered such that $\Delta_{\mathcal{E}}^{(1)} \leq \Delta_{\mathcal{E}}^{(2)} \leq ... \leq \Delta_{\mathcal{E}}^{(K)}$. With this we can estimate the log-likelihood function as[45]

$$\widehat{ll}(\boldsymbol{d}, \boldsymbol{s}; n, K) = -\sum_{\mathcal{E}} d_{\mathcal{E}} \log\left(s_n\left(\mathcal{D}_{\mathcal{E}}; \mathcal{D}_{\mathcal{E}}^{(1)}, ..., \mathcal{D}_{\mathcal{E}}^{(K)}\right)\right) + c(d_{\mathcal{E}})$$
(23)

where $c(d_{\mathcal{E}})$ is a constant, $d_{\mathcal{E}} = |\mathcal{I}_{\mathcal{E}}| + |\mathcal{P}_{\mathcal{E}}|$ is the dimensionality of data set $\mathcal{D}_{\mathcal{E}}$, and $s_n\left(\mathcal{D}_{\mathcal{E}}; \mathcal{D}_{\mathcal{E}}^{(1)}, ..., \mathcal{D}_{\mathcal{E}}^{(K)}\right) = \Delta_{\mathcal{E}}^{(n)}$ is the distance to the $n$th neighbor of $\mathcal{D}_{\mathcal{E}}$.

We optimize the approximate likelihood function in Eq. (23) with the CMA evolutionary strategy (CMA-ES)[46]. For optimization and evaluation we use the $n = 5$th nearest neighbor. Because the variance of

the approximate log-likelihood function in Eq. (23) decreases with increasing $K$, the noise handling strategy of CMA-ES selects $K \in \{5, 6, ..., 500\}$ adaptively, such that $K$ is small in regions where the direction of improvement in the approximate log-likelihood is obvious. This adaptivity saves computation time. To compute the performance $\Delta \log \hat{p}$ in Figs. 2 and S2 and S3, we compute $\log \hat{p}$ 10 times with $K = 10^4$ and $n = 5$ and take the average performance; the standard error of the mean is below 1.5 for all models and experiments, which is too small to be seen in the figures.

### Reporting summary
Further information on research design is available in the Nature Portfolio Reporting Summary linked to this article.

## Data availability
Source data for all figures are provided in the file 'source_data.zip' with this paper. The behavioral data extracted from published articles is available at https://github.com/jbrea/FoodCachingExperiments.jl.

## Code availability
The source code is available at https://github.com/jbrea/FoodCaching[47].

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

## Acknowledgements

This work was supported by the Swiss National Science Foundation with grants 200020_165538 (J.B. and W.G.), 200020_184615 (J.B. and W.G.) and 200020_207426 (J.B. and W.G.). J.B. thanks Ljerka Ostojić, Edward Legg, Piero Amodio, Ben Farrar, Vasiliki Liakoni, Samuel Muscinelli and Valentin Schmutz for helpful discussions and feedback. Special thanks go to Alireza Modirshanechi for important suggestions regarding likelihood-free inference and model comparison and to Anthony Dickinson for inspiring discussions at the beginning of the project and pointers to relevant literature.

## Author contributions

J.B., N.S.C., and W.G. designed the study. J.B. developed the models, wrote the code, and ran the simulations. J.B., N.S.C., and W.G. wrote the text.

## Competing interests

The authors declare no competing interests.
