## [Peer Review File · Nature Communications]

Computational models of episodic-like memory in food-caching birdsREVIEWER COMMENTS

Reviewer #1 (Remarks to the Author):

This is a heroic modelling paper which aims to build a model that can replicate the behaviour of food-hoarding jays across a total of 28 behavioural experiments conducted in Nicky Clayton's research group over the years. The main finding is that a model that uses model-free reinforcement learning ("Plastic caching") to explain future planning experiments, and therefore does not require mental time travel in the sense of playing back and searching through previous experiences.

Generally, this is a very interesting paper. I have a few questions about the assumptions and model building, which would be good to see explained more explicitly in the paper, however.

1) The main take-home message is based on the comparison between the model-free reinforcement model and the "Experience replay" model, which is supposed to represent mental time travel. It feels to me that proponents of a mental time travel strategy could always say that they disagree with how the mental time travel has been implemented in the model, since it was essentially there as the "straw man" to be shot down. However, the "experience replay" model is not crucial for the argument. The fact that the "plastic caching" model can recreate the outcomes of the future planning experiments without an explicit mental time travel strategy is the important outcome, whether or not an explicit mental time travel model can or cannot also recreate those outcomes.

2) The aspect of the model that worried me a little bit more is the optimization of "individual bird" parameters to each separate experiment. The authors acknowledge this, and present a version of the model where the same parameters have been fit to all the models, and indeed, such a "universal" model does not perform as well as the individually optimized models (as would be expected). I feel that this might be explained more explicitly in the main text. I also think that the authors might have attempted an in-between strategy, where the models were fit separately for the two species used in the actual experiments, as it is reasonable to assume that scrub jays and Eurasian jays do not have the same neural implementations of their caching modules.

3) I then have some specific questions about the decisions made about particular ways of implementing aspects of the model:

3.1) The "when" memory is modelled as a moving layer of weights that moves essentially to a different layer on each day (although it starts covering more than one day once it's past a certain number of days). Why did the authors implement time in that way, and more importantly, how realistic is that? It is

also unclear what happens to the weights in layer 5 as it accumulates weights from layer 4 over 3 days: are those weights added? Averaged? I realize it works, but it only works for the time intervals used in the behaviour experiments, which in turn of course were partially driven by the practicalities of running the experiments. One consequence of this way of implementing “when” is that it means animals learn about a certain interval (e.g. goes bad at 3 days), but this does not generalize to (say) 4 days. On the other hand, that may be realistic, as in a similar experiment in magpies, Zinkivskay et al (2009) trained the birds to retrieve one colour for one interval, and the other colour for another interval, which is consistent with the current models.

3.2) The weights of the connections between foods/locations and caching are separate from those between foods/locations and retrieval (“red” and “blue” modules). It would be interesting to know why this was implemented that way. In principle, it could be possible that the same memory structure controls both caching and retrieval. Was this because the retrieval module required the “when” component and the caching module did not?

3.3) Related to 3.2, the effects of not finding any food and finding “bad” food seem to have different effects: “no food” only affects the caching weights (Fig 4), while “bad food” affects both caching and retrieval weights (Fig. 3). In the actual model description (line 112), it is unclear (to me) whether “non-fresh” food includes “no food”, or only explicitly “food present, but not fresh”. Please clarify. Does “non-fresh” food only change $v_f(l)$, or does it also change $w(fx)_{cache}$?

3.4) Specific satiety is coded into the model by essentially having separate stomachs (with separate maximum capacities) for each food type. This is of course not realistic, as a stomach full of one food cannot really then be filled with a whole stomach’s worth of another food (although I acknowledge that birds who refuse to eat more of the first food might eat some of the second food). Does this unrealistic assumption affect the model in any significant way?

3.5) Similarly, the relationship between stomach content and hunger/satiety is unrealistic. Hunger continues to go down even as the stomach empties, and only starts to increase once the stomach is empty. Would changing this assumption affect the outcomes of the model in any way?

Finally, some suggestions for making the paper clearer:

a. The neohebbian rule is mentioned already on line 120, but not really explained until line 231. Can it be explained earlier for the uninitiated reader?

- b. The figures are much too small. The “wiggly grey arrows” are only visible when you blow up the figure to at least 200%.
- c. please also put all the parameters on all the panels in each figure, as it is not easy to have to refer back to panel A even when reading the description for panel C (e.g.).
- d. The numbering of the main figures seems to have changed at some point, because the Model Description refers to Figure 2A, when it probably means to refer to Figure 1C
- e. line 146 in the Model Description mentions a “gated synaptic term” – what is that, and to which term of Equation 12 does that refer?
- f. line 225 of Model Comparison and Fitting: there is an extra “and” in that sentence.
- g. Supplementary text: in Figure 2 caption, there is reference to a Figure 2E. Where is that?
- h. For all the outcomes of each experiment in the Supplementary materials, it would be good for each experiment to label the Y axes and explain any abbreviations used. Also, for 2 of the studies, it says “Published in [?]” Please fix.

Reviewer #2 (Remarks to the Author):

This is a very interesting and important work that presents a computational model of food caching and directly compares results from numerous experiments to two different cognitive processes – simple associative learning model and a higher level ‘mental time travel’ process.

Understanding whether non-human animals are capable of higher level cognition, similar to humans, such as mental time travel and episodic like memory is an important question in psychology, animal behavior and evolutionary biology. Traditionally, psychologists argued that animals do not have such abilities and that animals, unlike humans, are ‘stuck in time’. This notion has been challenged by numerous experiments arguing that animals are not stuck in time and that they are capable of mental time travel and future planning. Many of such experiments used food-caching species as a model to test whether at least some caching events represent planning and whether food caching and especially cache retrieval, starting with a classic episodic-like memory study in scrub-jays.

The main and important message of this study is that results from almost all previous studies claiming higher cognitive abilities can be explained by basic associative learning. The authors used the data from many of these studies and show that computational model based on basic associative learning provides the same outcome as the model based on higher cognitive abilities (Planning-by-replay model) and these both can explain results from the experimental studies. Considering that the outcomes are the same from a more simple and a more complex models, we cannot conclude that these studies indeed

demonstrated higher levels cognition – it is still possible, but more work is needed. The MS provides ways to improve testing in future studies by showing how to design an experiment that would allow clear discrimination between simple associative learning and more complex cognition involving mental time travel.

Overall, I think that the MS provides an important contribution and would be of great interest to a wide range of behavioral sciences.

I have only two main suggestions/requests.

1. First, the authors include ‘hunger variables for motivational control’. I think it would be important to report results of the model without such variables and without specific hunger-based motivational control. The reason is that while most lab-based studies use food deprivation to motivate animals to cache/retrieve, in the wild, pretty much all long-term caching is independent of such motivation. Birds like nutcrackers and many other, cache mostly on a seasonal basis and mostly when food is superabundant – so these birds are never hungry when caching. Nutcrackers are in particular a good example as they cache mostly seeds of various pine species during the time when these seeds are superabundant. The entire evolution of food caching involves caching food when it is plentiful to use later when food is scarce. So in most food-caching species including corvids, motivation to cache may be mostly genetically based (innate) rather than just hunger-based. It is also true, that these species would also cache on a much shorter time scale in the winter and such short-term caching may be controlled by different motivation including hunger. I think the MS would be better if additional models are included that do not have hunger-based motivational control.

2. Time is the most important component of the mental time travel and episodic memory. It is clear that ‘what’ and ‘where’ could be achieved by basic associative learning. The question is how such associative learning can encode time. Any temporal component within a day can be handled by associative learning using internal biological clock as a cue, but what about time across days? The authors state that basic associative learning in their model can allow ‘a flexible readout of how long ago a caching event happened..’(In. 115). Since this is the most critical part (in my view) of the MS, I think the authors should expand the description and justification of how basic associative learning allows animals to ‘estimate’ time across multiple days. Currently, it is not quite clear (at least to me) and I think such more extensive justification of how basic associative learning can handle time across days would be highly beneficial to many readers who are not necessarily neurobiologists or computational biologists. The authors provide a citation (27) that is also a computational model, but it would be important to have empirical studies showing that this is actually happening. Or is it entirely theoretical concept? If this process is not confirmed by empirical data, it needs to be discussed here. As I said, this is, in my view, the most critical component of the argument here. If animals cannot estimate duration of time across multiple days, the more complex model will likely be the only one explaining the outcome of the empirical experiments.

Reviewer #3 (Remarks to the Author):

The food caching behaviour of scrub jays has informed a number of influential models of memory function. It has been argued that the patterns of food searching that are observed after food has been cached indicate that the birds are engaged in a form of “mental time travel” – that they are recalling the sequence of events that have occurred in a particular order. Here, however, Brea and colleagues argue that similar patterns of behaviour are predicted by the activity of networks of neurons that are governed by Hebbian-like associative plasticity rules and in which connections between units (neurons) are strengthened by reward experience. They therefore argue that mental time travel need not be invoked to explain the birds’ behaviours.

The study is potentially of interest to researchers in the field. Perhaps one important feature is that it might define a new benchmark for the type of test that might be needed to demonstrate that memory is more than just associative.

1. I have really just one concern but that is a central one. It seems that there is a key reason why the authors’ model predicts scrub jay performance on memory tasks that require a sense of the order of events in time. Rather than invoking mental time travel the authors rely on a very particular type of consolidation process described on lines 112-122 in the second part of paragraph 1 on page 5. When memories are first created they are stored in one layer of the network but then they are moved on to the next layer before a new memory is laid down in the next test. This process is repeated over time. Therefore, identifying the layer in which a memory is located provides information about when the event happened. In other words, if memory is found in the first layers of the network then it reflects an event that happened recently but if it is found in more distant levels of the network then the memory is for a more temporally distant memory. It is the movement of memories across synapses in this way that does the work of providing an account of how the system “knows” when something happened. I think that the implications of this are probably twofold. First, the importance of the particular form of consolidation process might be given more weight throughout the manuscript (for example in the Abstract and Discussion – it is mentioned briefly in the Abstract and perhaps not at all in the Discussion). Second, there needs to be some clarity about whether or not there is neurobiological evidence for the movement of memories across synapses in the manner that is envisaged. At the moment, no such evidence is mentioned. By contrast, it needs to be acknowledged that the the mental time travel models are broadly consistent with demonstrations of a well-known biological phenomenon -- hippocampal replay.

Intelligence of Food Caching: Modelling Memory Formation and Behavior

Response to the Reviewers

Johanni Brea,^{1*} Nicola S. Clayton,² Wulfram Gerstner¹

¹School of Computer and Communication Science & School of Life Science,
École Polytechnique Fédérale de Lausanne, Lausanne, Switzerland

²Department of Psychology, University of Cambridge, Cambridge, UK

*To whom correspondence should be addressed; E-mail: johanni.brea@epfl.ch.

We thank all reviewers for their positive, constructive and helpful feedback!

We first reply to issues raised by all three reviewers before we turn to a point-by-point reply for each reviewer separately.

Contents

1	Issues raised by all reviewers	2
2	Reproducibility	3
3	Reviewer #1	4
4	Reviewer #2	9
5	Reviewer #3	11

1 Issues raised by all reviewers

We thank all reviewers for their valuable questions related to the implementation of time information in our associative memory model! In summary, the questions are:

1. Are there empirical studies showing that this implementation of the what-where-when memory is actually happening in the bird brain?

This is an important question; thank you for raising it! We are not aware of any empirical study supporting or falsifying our implementation. We clarify this in the discussion:

[line: 225] Although there is evidence for sharp-wave ripples during sleep in the hippocampus of the food-caching bird species tufted titmice [34] and sharp-wave ripples are believed to be important for systems memory consolidation [35], future experiments are needed to determine if the time information is memorized with systems memory consolidation and an associative learning mechanism and, if so, which one of several possible associative learning mechanisms [33] is actually implemented in the brains of food-caching birds.

2. Why is the what-where-when memory implemented in this way and how realistic is this implementation?

In the absence of strong empirical support for a specific mechanism we chose this implementation, because it clearly illustrates the essential idea. We clarify this in the discussion:

[line: 220] The specific implementation with moving synaptic connections illustrates the essential idea of how the age of memories can be retrieved in an associative memory with systems memory consolidation, but other implementations are possible ([33], Methods).

In the Methods we added the following paragraph:

[line: 422] This associative memory system with systems consolidation is just one hypothesis of automatic processes that keep track of when an event was memorized [33]. One alternative is to grow connections to all layers of the memory network at the moment of storage, while maintaining a time-dependant activity pattern at retrieval through synaptic connections that disappear at different rates, e.g. the connections to the first layer could disappear after one day, whereas those to other layers disappear later. In this case, a young memory would be characterized by many neurons being active during recall and an old memory by few neurons being active during recall. Another alternative, without multiple layers

and systems consolidation, is to store the “what-where” information together with a “when” tag, like a time stamp, or a “context” tag that allows to reconstruct the “when” information. Implementing the “when” information with a time tag or the number of active neurons during recall has computational disadvantages to the sparse code of the memory network $M(t)$, because quickly learning flexible rules based on the what-where-when of recalled events is easiest with linear readout, when the input to the linear readout is sparse, ideally, one-hot coded [33]. But further experiments are needed to discover the actual implementation of the what-where-when memory in food caching birds.

Your valuable questions regarding the implementation of the what-where-when memory stimulated us to look at this issue in detail. The result is a text that we would like to polish a bit further before we submit it later to a specialized journal as a separate article on a taxonomy of memory systems that keep track of time. The current version of this new work is available on bioRxiv [33] and is cited in the discussion and the methods section.

2 Reproducibility

Since reproducibility of experimental results is an important question, we ran additional simulations with 10 times more birds than used in the experiments. With this high number of subjects, we found that our main models can almost perfectly reproduce the statistically significant experimental results. We think this is an interesting finding and included it therefore in the main text and in the new figure 2B.

[line: 73] To investigate reproducibility, we also ran simulations with 10 times more subjects than in the real experiments (Fig. 2B).

[line: 279] Although the Plastic Caching Model has all the features to reproduce the experimentally observed behavior, some simulated repetitions of the experiments fail to reach significance on the key statistical tests with the low number of subjects typically used in the experiments (Fig. 2A-B). This suggests an alternative explanation for the recent failure of reproducing the breakfast planning experiments with Canada jays [43]: the sample number $N_{\text{experiment}} = 6$ in this experiment may have simply been too small.

3 Reviewer #1

This is a heroic modelling paper which aims to build a model that can replicate the behaviour of food-hoarding jays across a total of 28 behavioural experiments conducted in Nicky Clayton’s research group over the years. The main finding is that a model that uses model-free reinforcement learning (“Plastic caching”) to explain future planning experiments, and therefore does not require mental time travel in the sense of playing back and searching through previous experiences.

Generally, this is a very interesting paper. I have a few questions about the assumptions and model building, which would be good to see explained more explicitly in the paper, however.

1. The main take-home message is based on the comparison between the model-free reinforcement model and the “Experience replay” model, which is supposed to represent mental time travel. It feels to me that proponents of a mental time travel strategy could always say that they disagree with how the mental time travel has been implemented in the model, since it was essentially there as the “straw man” to be shot down. However, the “experience replay” model is not crucial for the argument. The fact that the “plastic caching” model can recreate the outcomes of the future planning experiments without an explicit mental time travel strategy is the important outcome, whether or not an explicit mental time travel model can or cannot also recreate those outcomes.

We agree and we added the following sentence to the discussion:

[line: 286] Even though the Planning-By-Replay Model is just one model of mental time-travel and other mental time-travel models are conceivable, the important point is that the Plastic Caching Model can reproduce the outcomes of the planning experiments without an explicit mental time-travel strategy.

2. The aspect of the model that worried me a little bit more is the optimization of “individual bird” parameters to each separate experiment. The authors acknowledge this, and present a version of the model where the same parameters have been fit to all the models, and indeed, such a “universal” model does not perform as well as the individually optimized models (as would be expected). I feel that this might be explained more explicitly in the main text. I also think that the authors might have attempted an in-between strategy, where the models were fit separately for the two species used in the actual experiments, as it is reasonable to assume that scrub jays and Eurasian jays do not have the same neural implementations of their caching modules.

We modified the results section, such that the respective part reads now:

[line: 69] For each model and experiment, the hyperparameters ϑ are adjusted by likelihood-free inference (Fig. 1B, Methods). We also fitted each model to all experiments jointly, such that the model-specific hyperparameters ϑ are the same for all experiments, but the simple models we considered fail to capture the strong inter-experimental variability, which is potentially due to seasonal or other unobserved effects (Appendix).

Regarding an in-between strategy: despite differences between Eurasian jays and California scrub-jays, it is unclear whether variability in caching behavior should be attributed more to species differences than to unobserved effects. Given that only two of the 28 experiments were done with Eurasian jays instead of California scrub-jays we could not investigate this systematically. A joint fit of all 26 experiments with California-scrub jays does not differ much from the joint fit with all experiments and a joint fit of the two experiments with Eurasian jays does not differ much from separate fits of the two experiments.

3. I then have some specific questions about the decisions made about particular ways of implementing aspects of the model:

- (a) The “when” memory is modelled as a moving layer of weights that moves essentially to a different layer on each day (although it starts covering more than one day once it’s past a certain number of days). Why did the authors implement time in that way, and more importantly, how realistic is that?

These are important questions, raised also by the other reviewers. Please see Issues raised by all reviewers.

It is also unclear what happens to the weights in layer 5 as it accumulates weights from layer 4 over 3 days: are those weights added? Averaged?

In the experiments it never happens that the same weight between caching-tray-feature neurons and foodtype neurons is moved on subsequent days from layer 4 to 5, because the birds never cached on different days in the same caching tray. Our implementation, however, is such that memory traces would be replaced, if such an event happened. We added the following explanation to the Methods:

[line: 394] In the experiments the birds never cached on different days in the same caching tray. Also in the wild, caching new items at a site where some food items are already cached is unlikely, given that these birds are scatter hoarders. In our model, repeated caching at the same site on multiple days would lead to replacement of the old memory trace, e.g. caching peanuts at x on day 1 and caching again peanuts at x on day 3 would lead to the deletion of the weights targeting layer 3 and the growth of new weights to layer 1.

I realize it works, but it only works for the time intervals used in the behaviour experiments, which in turn of course were partially driven by the practicalities of running the experiments. One consequence of this way of implementing “when” is that it means animals learn about a certain interval (e.g. goes bad at 3 days), but this does not generalize to (say) 4 days. On the other hand, that may be realistic, as in a similar experiment in magpies, Zinkivskay etal (2009) trained the birds to retrieve one colour for one interval, and the other colour for another interval, which is consistent with the current models.

Indeed, the current model does not generalize experiences from day 3 to day 4. This is why we think experiments to probe the limits of learnability and generalization to untrained retention intervals are needed, as mentioned in the discussion.

Thank you for the relevant reference. We included it in the discussion:

[line: 233] This model is also consistent with experiments where magpies were trained to retrieve objects of one color for one retention interval and objects of another color for another retention interval [36].

- (b) The weights of the connections between foods/locations and caching are separate from those between foods/locations and retrieval (“red” and “blue” modules). It would be interesting to know why this was implemented that way. In principle, it could be possible that the same memory structure controls both caching and retrieval. Was this because the retrieval module required the “when” component and the caching module did not?

Yes, indeed. This is the reason. We added the following explanation to the discussion:

[line: 255] Our implementation of a what-where-when memory in the associative memory module allows simulated birds to learn, for example, that unripe berries cached at a warm place are palatable after a few days, or that little pieces of meat are better preserved at cold and dry places than at warm and humid places, if the warmth and humidity of a cache site are part of the perceived cache-site features. For the plastic caching module, time is irrelevant, because the birds only need to learn, for example, that little pieces of meat should preferably be cached at cold and dry places or pilfered sites should be avoided.

- (c) Related to item 3b, the effects of not finding any food and finding “bad” food seem to have different effects: “no food” only affects the caching weights (Fig 4), while “bad food” affects both caching and retrieval weights (Fig. 3). In the actual model description (line 112), it is unclear (to me) whether “non-fresh” food includes “no food”, or only explicitly

“food present, but not fresh”. Please clarify. Does “non-fresh” food only change $v_f^{(l)}$, or does it also change w_{fx}^{cache} ?

Excellent question and observation. Indeed, it also changes w_{fx}^{cache} , see Eq. 13 of Materials and Methods. We stress this now in the results section by modifying the following sentence:

[line: 165] Second, the retrieval attempts of caches cause synapses w^{cache} (Fig. 4B) to decrease for birds that found degraded food or were unsuccessful in finding cached food and to increase for successful birds via a neoHebbian plasticity rule [28].

- (d) Specific satiety is coded into the model by essentially having separate stomachs (with separate maximum capacities) for each food type. This is of course not realistic, as a stomach full of one food cannot really then be filled with a whole stomach’s worth of another food (although I acknowledge that birds who refuse to eat more of the first food might eat some of the second food). Does this unrealistic assumption affect the model in any significant way?

In the experiments, the food types and quantities were chosen such that birds satiated on one food clearly continued desiring and eating another food. In the methods section “2.1 Motivational control” we acknowledge that a model based on nutrient classes would be more realistic but also more difficult to fit with the available data.

[line: 356] Instead of one stomach and hunger variable per food type one could assign one variable per nutrient class, e.g. carbohydrates, fats, fiber, minerals, proteins, vitamins, and water. However, the mapping between food types and nutrient classes is non-trivial and in the experiments the food types and quantities were chosen such that the birds satiated on one food clearly continued desiring and eating another food. Therefore we work directly with food types.

- (e) Similarly, the relationship between stomach content and hunger/satiety is unrealistic. Hunger continues to go down even as the stomach empties, and only starts to increase once the stomach is empty. Would changing this assumption affect the outcomes of the model in any way?

The hunger variable in the motivational-control module does indeed go down as the stomach empties and only starts to increase when the stomach variable reaches zero. This models the delay between food intake and the feeling of being satiated, because food absorption is not immediate. The hunger variable should not be interpreted as the load or the emptiness of the stomach, but rather as the difference between the current load value and a threshold value at which the bird’s hunger feeling starts to increase again. Indeed, we can modify the model such that hunger increases again when the stomach variable reaches some arbitrary positive threshold value $s_0 > 0$ without a change in the results after model

optimization, because the fitted parameters would adjust to a non-zero threshold s_0 . What turns out to be important during data fitting is not whether there is or is not threshold, but whether there is or is not a delay between food intake and decrease of the hunger level, and our model accounts for this delay.

To clarify this point we added the following explanation to the Methods:

[line: 369] Whereas the stomach variable increases immediately with every eaten food item and decreases linearly as food is digested, hunger decreases slowly during digestion, because food absorption is not immediate and major hunger satiation signals arise from the gut [30]. The value “zero” of the stomach variable should not literally be understood as indicating a completely empty stomach; rather it is the emptiness level of the stomach at which a bird’s hunger feeling starts to increase again.

Finally, some suggestions for making the paper clearer:

1. The neohebbian rule is mentioned already on line 120, but not really explained until line 231. Can it be explained earlier for the uninitiated reader?
2. The figures are much too small. The “wiggly grey arrows” are only visible when you blow up the figure to at least 200
3. please also put all the parameters on all the panels in each figure, as it is not easy to have to refer back to panel A even when reading the description for panel C (e.g.).
4. The numbering of the main figures seems to have changed at some point, because the Model Description refers to Figure 2A, when it probably means to refer to Figure 1C
5. line 146 in the Model Description mentions a “gated synaptic term” – what is that, and to which term of Equation 12 does that refer?
6. line 225 of Model Comparison and Fitting: there is an extra “and” in that sentence.
7. Supplementary text: in Figure 2 caption, there is reference to a Figure 2E. Where is that?
8. For all the outcomes of each experiment in the Supplementary materials, it would be good for each experiment to label the Y axes and explain any abbreviations used. Also, for 2 of the studies, it says “Published in [?]” Please fix.

Thank you! We implemented all suggestions.

4 Reviewer #2

This is a very interesting and important work that presents a computational model of food caching and directly compares results from numerous experiments to two different cognitive processes – simple associative learning model and a higher level ‘mental time travel’ process.

Understanding whether non-human animals are capable of higher level cognition, similar to humans, such as mental time travel and episodic like memory is an important question in psychology, animal behavior and evolutionary biology. Traditionally, psychologists argued that animals do not have such abilities and that animals, unlike humans, are ‘stuck in time’. This notion has been challenged by numerous experiments arguing that animals are not stuck in time and that they are capable of mental time travel and future planning. Many of such experiments used food-caching species as a model to test whether at least some caching events represent planning and whether food caching and especially cache retrieval, starting with a classic episodic-like memory study in scrub-jays.

The main and important message of this study is that results from almost all previous studies claiming higher cognitive abilities can be explained by basic associative learning. The authors used the data from many of these studies and show that computational model based on basic associative learning provides the same outcome as the model based on higher cognitive abilities (Planning-by-replay model) and these both can explain results from the experimental studies. Considering that the outcomes are the same from a more simple and a more complex models, we cannot conclude that these studies indeed demonstrated higher levels cognition – it is still possible, but more work is needed. The MS provides ways to improve testing in future studies by showing how to design an experiment that would allow clear discrimination between simple associative learning and more complex cognition involving mental time travel.

Overall, I think that the MS provides an important contribution and would be of great interest to a wide range of behavioral sciences.

I have only two main suggestions/requests.

1. First, the authors include “hunger variables for motivational control”. I think it would be important to report results of the model without such variables and without specific hunger-based motivational control. The reason is that while most lab-based studies use food deprivation to motivate animals to cache/retrieve, in the wild, pretty much all long-term caching is independent of such motivation. Birds like nutcrackers and many other, cache mostly on a seasonal basis and mostly when food is superabundant – so these birds are never hungry when caching. Nutcrackers are in particular a good example as they cache mostly seeds of various pine species during the time when these seeds are superabundant. The entire evolution of food caching involves

caching food when it is plentiful to use later when food is scarce. So in most food-caching species including corvids, motivation to cache may be mostly genetically based (innate) rather than just hunger-based. It is also true, that these species would also cache on a much shorter time scale in the winter and such short-term caching may be controlled by different motivation including hunger. I think the MS would be better if additional models are included that do not have hunger-based motivational control.

This is a good point. We ran additional simulations and report them in the new supplementary figure S5 and in the text as follows:

[line: 99] Whereas the motivation for eating depends undeniably on satiety, the caching behavior may be independent of hunger, as jays are known to cache predominantly in seasons when food is abundant [2]. Therefore, we compared the motivational control module to alternative motivational control models where caching is independent of the recent caching and eating history (unmodulated caching) or where caching depends only on the recent caching history (caching-modulated caching; Methods). We found that models without any motivational control reproduce the five specific satiety experiments clearly worse than models with hunger-modulated caching, but models with caching-modulated caching perform similarly to hunger-modulated caching models (Supplementary Fig. S5).

[line: 377] In the caching-modulated caching model of motivational control we use additionally the caching motivation variables c_f that evolve according to

$$\tau_d \frac{dc_f}{dt} = 1 - c_f(t) - c_f(t)c_0 \sum_k \delta(t - t_k) [a_k = \text{cache}_f] \quad (1)$$

where c_0 is a fitted parameter that controls how strongly the caching motivation variables c_f decreases when caching an item of type f .

[line: 458] In the unmodulated caching model we drop the term $v_f^{\text{cache}} h_f(t)$ in equation above and in the caching-modulated caching model we replace it by $v_f^{\text{cache}} c_f(t)$ (see also Supplementary Fig. S5).

2. Time is the most important component of the mental time travel and episodic memory. It is clear that ‘what’ and ‘where’ could be achieved by basic associative learning. The question is how such associative learning can encode time. Any temporal component within a day can be handled by associative learning using internal biological clock as a cue, but what about time across days? The authors state that basic associative learning in their model can allow ‘a flexible readout of how long ago a caching event happened..’(ln. 115). Since this is the most critical part (in my view) of the MS, I think the authors

should expand the description and justification of how basic associative learning allows animals to ‘estimate’ time across multiple days. Currently, it is not quite clear (at least to me) and I think such more extensive justification of how basic associative learning can handle time across days would be highly beneficial to many readers who are not necessarily neurobiologists or computational biologists. The authors provide a citation (27) that is also a computational model, but it would be important to have empirical studies showing that this is actually happening. Or is it entirely theoretical concept? If this process is not confirmed by empirical data, it needs to be discussed here. As I said, this is, in my view, the most critical component of the argument here. If animals cannot estimate duration of time across multiple days, the more complex model will likely be the only one explaining the outcome of the empirical experiments.

These are important questions and concerns, raised also by the other reviewers. To answer these question we added two paragraphs in the main text and a paragraph in the methods section. Please see Issues raised by all reviewers.

5 Reviewer #3

The food caching behaviour of scrub jays has informed a number of influential models of memory function. It has been argued that the patterns of food searching that are observed after food has been cached indicate that the birds are engaged in a form of “mental time travel” – that they are recalling the sequence of events that have occurred in a particular order. Here, however, Brea and colleagues argue that similar patterns of behaviour are predicted by the activity of networks of neurons that are governed by Hebbian-like associative plasticity rules and in which connections between units (neurons) are strengthened by reward experience. They therefore argue that mental time travel need not be invoked to explain the birds’ behaviours.

The study is potentially of interest to researchers in the field. Perhaps one important feature is that it might define a new benchmark for the type of test that might be needed to demonstrate that memory is more than just associative.

I have really just one concern but that is a central one. It seems that there is a key reason why the authors’ model predicts scrub jay performance on memory tasks that require a sense of the order of events in time. Rather than invoking mental time travel the authors rely on a very particular type of consolidation process described on lines 112-122 in the second part of paragraph 1 on page 5. When memories are first created they are stored in one layer of the network but then they are moved on to the next layer before a new memory is laid down in the next test. This process is repeated over time. Therefore, identifying the layer in which a memory is located provides information about when the event happened. In other words, if memory is found in the first layers of the network then it reflects an event that happened recently but if it is found in more distant levels of the network then the

memory is for a more temporally distant memory. It is the movement of memories across synapses in this way that does the work of providing an account of how the system “knows” when something happened. I think that the implications of this are probably twofold. First, the importance of the particular form of consolidation process might be given more weight throughout the manuscript (for example in the Abstract and Discussion – it is mentioned briefly in the Abstract and perhaps not at all in the Discussion).

This is an important point, indeed. To address it, we added two paragraphs in the maintext and one in the methods. Please see Issues raised by all reviewers.

Second, there needs to be some clarity about whether or not there is neurobiological evidence for the movement of memories across synapses in the manner that is envisaged. At the moment, no such evidence is mentioned.

Currently, our model is a hypothesis. We are not aware of neurobiological evidence in corvids that supports or falsifies our hypothesis. Please see Issues raised by all reviewers.

By contrast, it needs to be acknowledged that the mental time travel models are broadly consistent with demonstrations of a well-known biological phenomenon – hippocampal replay.

Good point! We extended the discussion on hippocampal replay as follows:

[line: 176] Despite similarities with hippocampal replay [17, 29], which would be consistent with the replay-and-plan module and models of mental-time-travel in general, we have not yet found a simple implementation of the replay-and-plan module (orange box in Fig. 4C) in terms of neural network dynamics and plasticity rules. In fact, a precise hypothesis of neural implementations of mental-time travel requires much more than hippocampal replay, as it would have to specify which aspects of the detailed multi-sensory processing stream are stored in the hippocampal replay memory, how the memory system can efficiently be queried, and how the outcome of multiple replayed episodes are combined to reach a decision for the next action (Methods).

[line: 521] The algorithmic description of the Planning-By-Replay Model illustrates three problems that a neural implementation of mental-time travel for planning needs to solve: first, what is stored in memory (Equation 15), second, how is the memory queried (Equation 16); and third, how are replayed episodes used for decision making (Equation 17). If food caching birds use indeed mental-time travel for planning, they probably do not store in memory every single observation and action, as if everything was recorded on a videotape. Instead they may store some compressed representation of past experiences, like the available food

or the average hunger level in the afternoon of a given day. Fast hippocampal replay of recent experiences may contribute to storing such compressed representations in long-term memory, but we do not yet have a detailed hypothesis of the underlying neural processes. Likewise, querying the memory system would involve probably non-trivial neural processing. The sight of a peanut should not trigger the memory system to retrieve the myriad of past experiences with peanuts. Instead, the memory system should be queried with information that is relevant in the given context. Finally, if a query leads to recall of multiple past episodes, their content should probably be held in some working memory for further information processing and decision making, which is likely to involve intricate neural processing.

REVIEWERS' COMMENTS

Reviewer #1 (Remarks to the Author):

I would like to thank the authors for answering all my questions and clarifying things further in the text.

Based on the clarifications and the extra models that have been run, I have a few more questions:

1. On line 116, you state that models with caching-modulated caching perform similarly to hunger-modulated caching models. You then refer to Supplementary Fig. S5. However, in S5, it seems that the Correia07 and Cheke11 experiments are equally well replicated in all models (even the unmodulated caching ones) and Clayton99C Experiments 2 and 3 never are, even with the caching-modulated caching. Only Experiment 1 from that paper changes with the addition of caching-modulated caching (over unmodulated). In S5's figure caption, you say you are "convinced" that a slightly more sophisticated way of making caching dependent on previous caching would get the same results as hunger-modulated caching, but that is not a sufficient basis for the statement on line 116 of the main paper.

2. Speaking of the caching-modulated caching - I don't understand the equation (4) that describes how the caching motivation variable c_f changes. What is c_0 ? Can you explain this a bit better for less computationally-inclined people, please?

3. Fig 2 now has an addition of running the experiments with 10 times more birds. The pattern of the graphs is the same as for Fig. 2A, but the Y axis has changed. Can you please put A and B on the same Axis, so the shift from near 50% to near 90% is more obvious?

4. Finally, I noticed a few editing mistakes (by no means exhaustive):

line 18: Corvidae should have a capital letter

line 275: birds' brains instead of birds' brain

line 366: characterized instead of charaterised

Note : all my line numbers are from the changes tracked version of the merged manuscript.

Reviewer #2 (Remarks to the Author):

The authors have done a great job addressing all comments/concerns. I have no additional comments.

Reviewer #3 (Remarks to the Author):

I think that the authors have addressed my main concerns. The discussion of their model in the context of other models and explanations is now more balanced and I think that, as currently formulated, it is more likely to draw positive interest from many readers.

Computational models of episodic-like memory in food-caching birds

Response to the Reviewers

Johanni Brea,^{1,2*} Nicola S. Clayton,³ Wulfram Gerstner^{1,2}

¹School of Computer and Communication Science,
École Polytechnique Fédérale de Lausanne, Lausanne, Switzerland

²School of Life Science,

École Polytechnique Fédérale de Lausanne, Lausanne, Switzerland

³Department of Psychology, University of Cambridge, Cambridge, UK

*To whom correspondence should be addressed; E-mail: johanni.brea@epfl.ch.

We thank reviewer #1 for the additional questions!

1. On line 116, you state that models with caching-modulated caching perform similarly to hunger-modulated caching models. You then refer to Supplementary Fig. S5. However, in S5, it seems that the Correia07 and Cheke11 experiments are equally well replicated in all models (even the unmodulated caching ones) and Clayton99C Experiments 2 and 3 never are, even with the caching-modulated caching. Only Experiment 1 from that paper changes with the addition of caching-modulated caching (over un-modulated). In S5's figure caption, you say you are "convinced" that a slightly more sophisticated way of making caching dependent on previous caching would get the same results as hunger-modulated caching, but that is not a sufficient basis for the statement on line 116 of the main paper.

Thank you for raising this point. It turns out that a more sophisticated filter allows to reproduce one of the three experiments that were not perfectly reproducible with caching-modulated caching, but not the other two. We changed the main text to

[line: 103] We found that models without any motivational control reproduce the five specific satiety experiments clearly

worse than models with hunger-modulated caching, whereas models with caching-modulated caching perform better than hunger-modulated caching models on one experiment (Clayton99C exp1) and worse on two experiments (Clayton99C exp3 and Cheke11 specsat; see Supplementary Fig. S5).

and we added panel C in Fig S5. and changed the caption to

In experiments Clayton99C exp3 and Cheke11 specsat, birds that ate to satiety on uncacheable, powdered food, cached subsequently fewer cacheable items of the same kind of food than birds under control conditions. This behavior is not reproducible with caching-modulated caching. **C** If caching-modulated caching is modelled with filters of the same kind as the hunger variables (instead of a simple low-pass filter), the results of Clayton99C exp2 can also be reproduced, but the experiments Clayton99C exp3 and Cheke11 specsat remain unreproducible.

2. Speaking of the caching-modulated caching – I don't understand the equation (4) that describes how the caching motivation variable c_f changes. What is c_0 ? Can you explain this a bit better for less computationally-inclined people, please?

Yes, we added the following explanation:

[line: 382] In other words, whenever a simulated bird caches an item of type f , the caching motivation c_f for food f decreases by the amount $c_f(t) \cdot c_0/\tau_d$ towards zero and in the absence of caching events it increases exponentially to one with time-constant τ_d . If the fitted parameter c_0 is large, few caching events suffice to bring the caching motivation close to zero.

3. Fig 2 now has an addition of running the experiments with 10 times more birds. The pattern of the graphs is the same as for Fig. 2A, but the Y axis has changed. Can you please put A and B on the same Axis, so the shift from near 50% to near 90% is more obvious?

Good suggestion, thank you.

4. Finally, I noticed a few editing mistakes (by no means exhaustive):

Thank you! We corrected these and other typos.